# Vaccination Strategies Against Hepatic Diseases: A Scoping Review

**DOI:** 10.3390/vaccines14010049

**Published:** 2025-12-31

**Authors:** Zahra Beyzaei, Bita Geramizadeh, Sara Karimzadeh, Ralf Weiskirchen

**Affiliations:** 1Transplant Research Center, Shiraz University of Medical Sciences, Shiraz 7193711351, Iran; geramib@gmail.com; 2Department of Pathology, Medical School of Shiraz University, Shiraz University of Medical Sciences, Shiraz 7193711351, Iran; 3Shiraz Medical School Library, Shiraz University of Medical Sciences, Shiraz 7193711114, Iran; karimzadeh_2860@yahoo.com; 4Institute of Molecular Pathobiochemistry, Experimental Gene Therapy and Clinical Chemistry (IFMPEGKC) RWTH University Hospital Aachen, D-52074 Aachen, Germany

**Keywords:** vaccination strategies, hepatic diseases, liver immunization, viral hepatitis prevention, vaccine efficacy

## Abstract

**Background/Objectives**: Viral hepatitis remains a significant global cause of chronic liver disease, highlighting the importance of effective vaccination strategies. This review assesses recent evidence on vaccine safety and effectiveness. **Methods**: A comprehensive search of PubMed, Embase, Web of Science, and Scopus identified English-language studies published from January 2000 to September 2025. Eligible studies evaluated vaccination for hepatitis A, B, C, or E, as well as vaccine responses in individuals with chronic liver disease or HIV infection. Of 5254 records screened, 166 studies met the inclusion criteria. **Results**: Hepatitis A vaccines demonstrated excellent safety, 95–100% short-term seroprotection, and durable immunity for both inactivated and live-attenuated formulations, with population-level reductions in disease incidence. Hepatitis B vaccines showed consistently strong immunogenicity across age groups, with over 90% seroprotection from recombinant and CpG-adjuvanted formulations. Effective prevention of mother-to-child transmission required maternal antiviral therapy, timely birth-dose vaccination, hepatitis B immunoglobulin (HBIG) administration, and post-vaccination serologic testing. Long-term data demonstrated immune persistence for up to 35 years and significant reductions in liver cancer following neonatal HBV vaccination. Limited studies in hepatitis C populations showed impaired responses, partially improved with higher or booster doses. Hepatitis E vaccines showed excellent safety and over 99% seroconversion. In non-viral liver disease and post-transplant populations, vaccine responses were reduced but remained clinically meaningful, especially with adjuvanted or higher-dose HBV vaccines. Among HIV-infected individuals, HAV vaccination was generally effective, while enhanced HBV regimens markedly improved seroprotection. **Conclusions**: Hepatitis A, B, and E vaccines are safe, immunogenic, and effective, with neonatal hepatitis B vaccination critical for preventing maternal transmission. No licensed HCV vaccine exists, and therapeutic HCV vaccines show limited efficacy. Optimized and targeted vaccination strategies are needed for individuals with chronic liver disease, HIV infection, HCV infection, transplant recipients, and other immunocompromised populations to maximize public health impact.

## 1. Introduction

Vaccination remains one of the most effective interventions in global health, serving as a cornerstone for the prevention and control of infectious diseases [1]. By stimulating adaptive immunity and providing herd protection, vaccines have significantly decreased the global burden of many viral infections, leading to substantial improvements in life expectancy and quality of life. However, liver-related infectious diseases continue to have a disproportionate impact on health and the economy, especially in low- and middle-income regions with limited access to preventive measures and healthcare infrastructure [2].

Hepatic diseases encompass a wide range of disorders, with viral hepatitis being a primary cause of chronic liver disease, cirrhosis, and hepatocellular carcinoma (HCC) [2,3]. Viral hepatitis remains a significant global health threat and a major contributor to mortality worldwide. The five main hepatotropic viruses, hepatitis A virus (HAV), hepatitis B virus (HBV), hepatitis C virus (HCV), hepatitis D virus (HDV), and hepatitis E virus (HEV), vary in their transmission modes, geographic distribution, viral structure, clinical outcomes, likelihood of inducing liver-related complications, and response to available treatments. The global prevalence of viral hepatitis remains high and is increasing over time. Viral hepatitis can lead to liver inflammation, which may progress to fibrosis, cirrhosis, and HCC over time. Chronic infections with HBV and HCV are associated with an increased risk of cancer, primarily due to the ongoing inflammatory response they cause in the liver [4].

The World Health Organization estimates that approximately 354 million individuals are chronically infected with hepatitis B or C viruses, resulting in more than 1.1 million deaths annually [5]. While the introduction of hepatitis A and B vaccines has led to remarkable declines in infection rates and liver-related complications in many regions, persistent challenges, such as incomplete vaccine coverage, emerging viral genotypes, and the absence of effective vaccines for hepatitis C, and limited availability of hepatitis E vaccines, continue to hinder global elimination goals [6,7]. In addition, vaccine hesitancy, socioeconomic disparities, and limited public health infrastructure further compromise the success of immunization programs targeting hepatic pathogens. Notably, hepatitis B vaccination also provides indirect protection against HDV, highlighting the broader benefits of universal HBV immunization.

Against this background, vaccination strategies aimed at hepatic diseases deserve renewed attention as a crucial component of liver disease prevention and global health policy. Advances in molecular immunology, vector-based platforms, and mRNA technology are opening up new avenues for hepatic vaccine research, promising broader protection and improved efficacy [8,9].

This scoping review aims to offer a comprehensive assessment of vaccination strategies targeting hepatic diseases, with a focus on their immunological mechanisms, effectiveness, and impact on the burden of liver disease. By incorporating current evidence from clinical and epidemiological studies, this review aims to highlight the essential role of vaccination in hepatology and to pinpoint challenges and opportunities for optimizing vaccine-based interventions in reducing hepatic morbidity and mortality.

## 2. Materials and Methods

This scoping review synthesizes current evidence on vaccination strategies among patients with hepatic diseases, with the primary aim of mapping the extent, characteristics, and nature of the available literature rather than evaluating comparative effectiveness or clinical superiority. No formal hypothesis testing or quantitative synthesis was planned or conducted. The reporting of this review followed the guidelines of the Preferred Reporting Items for Systematic Reviews and Meta-Analyses extension for Scoping Reviews (PRISMA-ScR) [10]. In accordance with scoping review methodology, no formal risk-of-bias assessment, quality appraisal, or certainty grading (e.g., GRADE) was performed. The review protocol was registered in PROSPERO (registration no. CRD420251249151). A completed PRISMA checklist is provided in the Appendix A.

A thorough literature search was conducted on PubMed, Embase, Web of Science, Scopus, and ClinicalTrials.gov to identify English-language studies published from January 2000 to September 2025. A medical librarian with expertise in systematic searching (SK) developed the search strategy using a combination of subject headings and keywords. The search included relevant keywords and Medical Subject Headings (MeSH) related to: “Liver Diseases,” “Viral Hepatitis,” “Hepatitis A,” “Hepatitis B,” “Hepatitis C,” “Alcoholic Liver Disease*,” “Alcoholic Fatty Liver,” “Alcoholic Hepatitis,” “Alcoholic Liver Cirrhosis,” “Fibrosis,” “Non-alcoholic Fatty Liver Disease,” “Liver Failure,” “Liver Neoplasms,” “Hepatocellular Carcinoma*” AND “Vaccination Methods,” “Hepatitis A Vaccines,” “Hepatitis B Vaccines,” “Hepatitis C Vaccines,” “Hepatitis E Vaccines,” “Viral Hepatitis Vaccines.” All search strategies are available in the Appendix A.

Studies were included if they primarily evaluated the immunogenicity, safety, or effectiveness of hepatitis vaccines in human populations, including individuals with hepatic diseases or related risk conditions. Studies assessing vaccines in people living with HIV (PLWH) were included when the effects of vaccination could be reasonably assessed independently of concurrent antiviral therapy. Exclusion criteria included: 1. Antiviral therapy as the primary intervention without a focus on hepatitis vaccination; 2. Non-English studies; 3. Studies not reporting primary or secondary data (e.g., editorials, opinion pieces, and review articles); 4. Studies focused on vaccines in populations other than patients with hepatic diseases; 5. Non-human studies, case reports, and conference abstracts.

Data were obtained from a single dataset in the original study if multiple centers were reported; however, when multiple datasets were presented, only one complete and non-overlapping dataset was extracted to avoid duplication. We requested the full text from the corresponding author, and studies for which no response was received were excluded from the review.

All references were managed using EndNote (version 20.4.1). Duplicate records were identified and excluded, leaving the remaining articles to be screened based on their title and abstract. Full-text evaluation was then conducted to determine their eligibility. Subsequently, the reference lists of all included studies were reviewed to identify any additional relevant publications. Four investigators (ZB, BG, SK, RW) screened the studies after an initial team meeting to discuss the purpose of the review and define inclusion and exclusion criteria. Two authors (ZB, RW) screened the full texts, and any disagreements in screening were resolved through consensus discussion.

## 3. Results

In total, 166 studies were identified, all published between 2000 and 2025 (Figure 1). Of these, 33 focused on Hepatitis A vaccination, 64 on Hepatitis B vaccination, five on Hepatitis C vaccination, five on Hepatitis E vaccination, 18 on vaccination for non-viral hepatitis, and 41 on hepatitis vaccination in patients with HIV. The studies included in this review were conducted across multiple global regions. The largest proportion originated from China (50 studies, 28%), followed by Europe (35 studies, 20%) and the United States (19 studies, 11%). The remaining studies were distributed across other parts of Africa, Asia, and South America. Fifty-seven publications (34%) employed a randomized controlled trial (RCT) design, 51 (31%) used a cohort design, and 25 (15%) reported cross-sectional data. Seventeen studies (10%) were observational, and the remaining 15 (10%) used other study designs.

### 3.1. Vaccines for Viral Hepatitis A

A total of 33 studies [11,12,13,14,15,16,17,18,19,20,21,22,23,24,25,26,27,28,29,30,31,32,33,34,35,36,37,38,39,40,41,42,43] from Asia, Europe, North and South America were identified, including RCTs, cohort studies, cross-sectional analyses, and national observational cohorts. These studies included 27,893 participants across all age groups, from infants to adults, and encompassed diverse populations, including unique high-risk groups such as Alaska Native communities. The vaccine formulations evaluated included both inactivated hepatitis A vaccines (HAVac), such as Healive^®^, Havrix^®^, Avaxim™, Havisure™, and HAVpur Junior, as well as live-attenuated HAV vaccines (HAV-L) (Appendix A).

#### 3.1.1. Safety of Hepatitis A Vaccines

Across the included studies, HAVac was generally reported to have a favorable safety profiles across different populations, vaccine formulations, and study designs. The most commonly reported local adverse events were injection-site pain, swelling, and erythema, but these reactions were generally mild and temporary [14,15,16]. Systemic symptoms like headache, low-grade fever, and fatigue occurred infrequently and at similar rates across various vaccines, including Healive^®^, Havrix^®^, Avaxim™, Havisure™, HAVpur Junior, and HAPIBEV™ [12,15,30,34,40,42]. Importantly, no RCTs or observational studies reported any serious adverse events related to the vaccines. A study that assessed the coadministration of the HAVac with the tetravalent dengue vaccine TAK-003 also found no safety concerns or increased reactogenicity [13]. National surveillance studies from Turkey and Panama have shown significant decreases in hepatitis A hospitalizations without an increase in vaccine-related complications. Therefore, these data demonstrate that HAVac is safe for diverse populations, with a consistent and favorable tolerability profile.

#### 3.1.2. Immunogenicity and Short-Term Response to Hepatitis A Vaccination

Short-term immunogenicity outcomes were reported in multiple studies across all vaccine formulations and age groups, with most studies describing seroconversion and seroprotection rates between 95% and 100% after one or two doses [14,15,16,17,18,19]. Randomized trials comparing Healive^®^, Havrix^®^, Avaxim™, and other regional formulations like Havisure™, HAVpur Junior, and HAPIBEV™ were primarily designed as non-inferiority studies. Individual studies reported non-inferiority between products, with some studies describing significantly higher geometric mean concentrations (GMC) or titers in groups receiving Healive^®^ or Avaxim™ [28,30,31,32,33]. Live-attenuated vaccines achieved similarly high seroprotection rates. However, several studies noted that their early antibody titers were sometimes lower than those induced by inactivated vaccines [18,19,24,30,34,38]. Single-dose universal vaccination programs, as reported in studies conducted in Brazil and China, described strong serologic responses in young children, with seropositivity rates surpassing 90% [25,26,30,36]. The included studies reported high short-term immunogenicity outcomes following both inactivated HAVac and live-attenuated HAVac, with heterogeneity in study design, populations, and outcome measures.

#### 3.1.3. Long-Term Protection and Antibody Persistence Following Hepatitis A Vaccination

Long-term follow-up studies have shown persistent immunity after HAV vaccination. Ramaswamy et al., [17], Mosites et al., [23], and Spradling et al., [33] reported that protective anti-HAV antibody levels remained for 10–25 years, with models predicting persistence for ≥30 years in most participants. Theeten et al., [35] demonstrated that adults who received a two-dose inactivated HAVac remained largely seropositive for 20 years, with models predicting ≥95% persistence at 30 years, confirming the durable immunogenicity of the inactivated HAVac. Live-attenuated vaccines also showed exceptional stability, maintaining high seroprotection 15–17 years after a single childhood dose and showing strong anamnestic responses upon booster challenge [18]. Studies have reported the persistence of HAV-specific memory B and T cells many years after vaccination, suggesting the presence of immune memory even in cases where circulating antibody levels declined. In addition, levels of cytokines such as interleukin (IL)-6, IL-10, tumor necrosis factor (TNF), and interferon gamma (IFN-γ) were shown to increase following vaccination [24,36]. Together, these studies describe long-term immunological responses following both inactivated and live-attenuated HAV vaccination, as reported over extended follow-up periods.

#### 3.1.4. Comparative Performance Across Hepatitis A Vaccine Types

Across multiple trials, studies have reported that all HAVac are associated with high immunogenicity outcomes. Some studies have described differences in antibody concentrations or stability between formulations. Inactivated vaccines like Healive^®^ and Avaxim™ were reported in some studies to have higher antibody concentrations than Havrix^®^, while high seroprotection rates were described for all evaluated vaccines [16,20,30,42]. Newly developed or locally manufactured vaccines, such as Havisure™, HAPIBEV™, and HAVpur Junior, consistently demonstrate non-inferiority to established products, with comparable safety and immunogenicity profiles [14,15,40]. Live-attenuated vaccines, while sometimes resulting in lower early antibody titers, show excellent long-term persistence, often matching or surpassing the duration of immunity seen with inactivated vaccines [30,34]. Population-level analyses from Turkey and Panama further emphasize the real-world impact of HAVac, showing significant decreases in hepatitis A incidence and hospitalization after nationwide implementation [22,37]. Therefore, available evidence suggests that HAVacs are effective, with inactivated vaccines generally associated with early immunogenic responses and live-attenuated vaccines demonstrating sustained immune persistence over long-term follow-up.

### 3.2. Vaccines for Viral Hepatitis B

Hepatitis B vaccines (HBVacs) have been extensively evaluated in diverse populations, including healthy adults, children, high-risk groups, and pregnant women. A total of 64 studies [44,45,46,47,48,49,50,51,52,53,54,55,56,57,58,59,60,61,62,63,64,65,66,67,68,69,70,71,72,73,74,75,76,77,78,79,80,81,82,83,84,85,86,87,88,89,90,91,92,93,94,95,96,97,98,99,100,101,102,103,104,105,106,107] conducted in Asia, Europe, and North and South America were identified. These studies comprised RCTs, cohort studies, cross-sectional analyses, and national observational cohorts, including more than 250,000 participants spanning all age groups from infants to adults and representing a wide range of demographic and clinical settings. Globally, recombinant HBsAg-based vaccines, administered alone or with HBIG in perinatal settings, remain the standard of care. Various formulations, including single-antigen (1A-HBV), three-antigen (3A-HBV), alum-adjuvanted vaccines, and CpG-adjuvanted vaccines (e.g., Heplisav-B), induce robust seroprotection across diverse populations, as summarized in Appendix A.

#### 3.2.1. Safety of Hepatitis B Vaccines

HBVacs have demonstrated an excellent safety profile across various populations. Adverse events are typically mild, consisting of transient local or systemic reactions, with serious adverse events being rare [64,88,90]. Safety remains consistent among adults, adolescents, neonates, and immunocompromised groups, including pregnant women [49,59]. Both conventional and novel formulations, such as CpG-adjuvanted vaccines, are well-tolerated and suitable for widespread immunization programs.

#### 3.2.2. Immunogenicity and Short-Term Hepatitis B Vaccine Response

HBV vaccines have been reported to rapidly induce protective anti-HBs antibodies. Seroconversion rates after standard 3-dose schedules (0–1–6 months) typically exceed 95% in infants and children, and 90–100% in healthy adults [44,45,63,64]. CpG-adjuvanted vaccines (Heplisav-B) demonstrate slightly faster and higher antibody responses in adults with low baseline anti-HBs [45]. Booster doses in non-responders reliably induce protective immunity, with seroconversion rates of at least 80% [66,76,107]. The study further demonstrated immune persistence and booster efficacy in children across different pre-booster anti-HBs strata. Children with pre-booster anti-HBs concentrations of 1–<10 mIU/mL exhibited significantly stronger and more stable responses compared with those who had lower baseline titers (*p* < 0.001). Notably, robust antibody persistence was maintained for up to eight years following the booster dose [67]. Multiple studies reported seroconversion rates exceeding 90% in healthy adults and children [44,45,46,63,65,89,91,94,98,107]. In neonates born to HBsAg-positive mothers, combination prophylaxis with HBVac and HBIG effectively prevents mother-to-child transmission (PMTCT), even with high maternal viral loads [49,52,58,68,78]. Zhang et al. [81] reported an assessment of risk factors associated with hepatitis B breakthrough infection (HBBI) following vaccination, with particular focus on antibody response patterns. The overall HBBI rate was 5.36%, with the highest incidence observed among individuals aged 18–29 years (7.33%). Breakthrough infections varied significantly by township and correlated with differences in anti-HBs responses. Notably, hyporesponsiveness to the HepB vaccine was identified as an independent predictor of HBBI. These findings underscore the importance of monitoring young adults for potential new HBV infections. Zhou et al. highlighted the immunological complexity underlying therapeutic HBV vaccination. In their RCTs of 93 chronic hepatitis B patients, individual cytokine or chemokine levels did not reliably predict hepatitis B e-Antigen (HBeAg) seroconversion after administration of the HBsAg–HBIG therapeutic vaccine (YIC). However, multivariate analysis of 14 cytokines/chemokines, particularly IL-10, IL-33, and macrophage inflammatory protein-1α (MIP-1α), offered improved discriminatory capacity (sensitivity 0.59, specificity 0.80), suggesting potential utility of composite biomarker panels for optimizing candidate selection in therapeutic vaccination [80].

In an RCT by Lian et al., involving 303 treatment-naïve HBeAg-positive chronic hepatitis B patients across nine liver centers in China, the efficacy and safety of a combination therapy including pegylated INF-α-2b, tenofovir disoproxil fumarate (TDF), granulocyte-macrophage colony-stimulating factor (GM-CSF), and HBVac were evaluated against standard treatments. HBsAg seroconversion at week 48 was observed in 3.0% of patients in the experimental group, compared to 1.03% in the TDF plus pegylated INF-α group and 1.19% in the IFN-alone group, with no statistically significant differences (*p* = 0.629). The decline in HBsAg levels was greater in both the experimental and TDF plus peg-IFN groups compared with IFN alone (*p* = 0.008 and 0.006, respectively), but there was no significant difference between the experimental and TDF plus peg-IFN groups (*p* = 0.619). Adverse events were similar across groups, with a lower incidence of neutropenia in the experimental group. Therefore, the combination therapy did not demonstrate superiority over TDF plus peg-IFN [53].

#### 3.2.3. Long-Term Protection and Antibody Persistence Following Hepatitis B Vaccination

Long-term follow-up studies show stable protection, often lasting 20–35 years. While circulating anti-HBs titers decrease, immune memory through B and T cells remains robust [55,75,102]. Breakthrough infections are rare and mostly asymptomatic, with rapid anamnestic responses observed after booster doses. Optimal scheduling and timely post-vaccination serologic testing (PVST) improve early detection of non-responders and ensure ongoing protection [57,62]. Vaccination triggers coordinated humoral and cellular responses, including increased levels of IL-6, IL-10, TNF-α, and IFN-γ, supporting short-term antibody production and immune memory establishment [44,51,75,76,87].

In a long-term RCT by Cao et al., 82,866 participants followed from infancy received HBVac at birth, 1 month, and 6 months. After 37 years of follow-up, vaccinated individuals had a significantly lower incidence of HCC compared to unvaccinated controls (Hazard Ratio (HR) 0.28; 95% Confidence Interval (CI) 0.11–0.70; *p* = 0.007). Importantly, the observed reduction in liver cancer incidence was attributable to a decrease in HBV-related HCC, consistent with the etiologic role of chronic hepatitis B infection in hepatocarcinogenesis. No protective effect was reported for HCC arising from non-viral causes. The efficacy of neonatal HBVac was 72% for preventing HBV-related HCC, 70% against liver cancer-related deaths, and 64% against deaths from liver disease, demonstrating strong and sustained long-term protection [54]. Neonatal vaccination also markedly reduced the incidence of primary HBV-related liver cancer (84%, 95% CI 23–97%), mortality from severe end-stage liver disease (70%, 95% CI 15–89%), and infant fulminant hepatitis (69%, 95% CI 34–85%). In contrast, catch-up vaccination at 10–14 years showed substantially lower effectiveness (21% vs. 72% for neonatal vaccination). Among children born to HBsAg-positive mothers, an adolescent booster dose (10–14 years) further reduced HBsAg prevalence (HR 0.68; 95% CI 0.47–0.97). Therefore, these findings underscore the critical importance of universal neonatal hepatitis B vaccination and targeted adolescent boosting in high-risk populations for the prevention of HBV-associated HCC [106].

Long-term follow-up studies have shown that despite a decrease in anti-HBs levels over the years, immune memory remains strong. In a study by Bruce et al., it was reported that 35 years after primary vaccination, 47.3% of participants still had protective anti-HBs levels, and a booster shot was able to restore seroprotection in most individuals with low titers [55]. Similarly, adolescents and adults who were vaccinated in infancy or childhood displayed robust anamnestic responses when challenged, indicating the stability of their immune memory [61,102].

#### 3.2.4. Prevention of HBV Mother-to-Child Transmission

Extensive evidence from diverse geographical regions supports the effectiveness of hepatitis B immunoprophylaxis strategies in PMTCT and improving post-vaccination immune responses. In a randomized comparative study from China, Wang et al. evaluated 331 mother–infant pairs and demonstrated that PMTCT rates did not differ significantly between infants receiving 100 IU versus 200 IU of HBIG in combination with HBVacs (*p* > 0.05). A high maternal HBV DNA level remained the main risk factor for immunoprophylaxis failure. Therefore, a single 100 IU HBIG dose combined with HepB vaccination was sufficient and more cost-efficient [49]. Similar findings were reported in a large Nigerian cohort by Ndububa et al., who screened 10,866 pregnant women, of whom 395 had chronic HBV infection. 376 mother–infant pairs completed follow-up. Among infants receiving HepB alone or HepB plus HBIG, alongside antiviral therapy for high-risk mothers, all infants tested HBsAg-negative at nine months, demonstrating complete prevention of PMTCT. Maternal HBV prevalence was 3.64%, and only 5.2% were HBeAg-positive [52]. In an observational study by Zhou et al., in China, 2120 HBsAg-positive mother–infant pairs were evaluated for PVST to prevent mother-to-child HBV transmission. The PVST participation rate was 67.1%. Among infants, 0.77% were HBsAg-positive, 96.8% were anti-HBs-positive, and 2.4% were both negative. Of the double-negative infants, 93.3% achieved antibody positivity after revaccination. Higher maternal education, older maternal age, and local residence were associated with greater PVST follow-up. These results indicate that PVST and timely revaccination effectively enhance protection against HBV infection in infants [57]. In a feasibility study in the Democratic Republic of the Congo, Thompson et al., enrolled 90 pregnant women through an HIV PMTCT program. Among 88 live-born infants, 68% received a birth-dose HBVacs, with 77% vaccinated within 24 h. Coverage was higher at study facilities compared with outside facilities, and no adverse events were reported. At 24 weeks, all tested infants were HBsAg-negative, demonstrating complete prevention of PMTCT [59]. Similarly, Miyakawa et al. in Vietnam followed 1339 children from 1987 mother–child pairs after universal vaccination introduction. 1.9% of children were HBsAg-positive at two years, with 28.3% infection among children of HBeAg-positive mothers despite full vaccination. Risk factors for infection or low antibody levels included low maternal body mass index (BMI) and incomplete HepB dosing, though the overall infection rate in children remained low [60]. Therefore, these studies suggest that birth-dose vaccination, PVST, and timely revaccination are associated with reduced rates of HBV PMTCT, with maternal and programmatic factors influencing reported outcomes.

The effect of vaccine dose on PMTCT outcomes was investigated in a multicenter cohort study by Zhang et al., which included 955 infants born to HBsAg-positive mothers. Although increasing the HepB dose from 10 µg to 20 µg did not significantly reduce MTCT, the higher dose enhanced immunogenicity, resulting in more infants with high anti-HBs titers (≥1000 IU/L) and fewer low responders, especially among those born to mothers with HBV DNA < 5 log10 IU/mL [58]. Wei et al. compared 100 IU versus 200 IU HBIG in 1177 infants born to HBsAg-positive mothers. PMTCT rates at seven and twelve months remained low and statistically similar between the groups (1.5% vs. 1.9%), while anti-HBs seropositivity exceeded 97% in both cohorts, supporting the adequacy of the 100 IU dose when combined with the standard HepB vaccine series [84]. Lu et al. assessed neonates born to HBsAg-positive/HBeAg-negative mothers and found PMTCT rates of 0.1% among those receiving vaccine + HBIG and 0.0% in infants receiving vaccine alone. Both groups achieved anti-HBs response rates over 97%, and vaccine-only recipients had higher geometric mean titers (GMT) at seven and twelve months. No cases of horizontal HBV transmission were detected, indicating that the vaccine alone is sufficient in mothers who are HBeAg-negative [97].

#### 3.2.5. Comparative Performance Across Hepatitis B Vaccine Types

Comparative studies have highlighted differences in immunogenicity among HBV vaccine formulations. Recipients of 3A-HBV achieve higher and more persistent seroprotection compared to 1A-HBV [46]. CpG-adjuvanted vaccines (Heplisav-B) provide stronger antibody responses than standard alum-adjuvanted vaccines, especially in adults with low pre-existing anti-HBs titers [45]. Increasing the dose in neonates (10 μg vs. 20 μg) improves antibody response and reduces occult HBV infection [58,68]. In an RCT in Bangladesh, Chowdhury et al. evaluated the immunogenicity and safety of a locally produced Hepa-B vaccine compared with Engerix-B in 158 healthy adults following a three-dose schedule (0, 1, and 6 months) [63]. Seroconversion rates were 98.7% for Hepa-B and 100% for Engerix-B, with GMT ratios exceeding the 0.5 non-inferiority margin at all time points. These results demonstrate that Hepa-B is non-inferior to Engerix-B in both immunogenicity and safety [63]. Standard recombinant HBV vaccines remain effective in most populations, with boosters reserved for high-risk groups or non-responders [98,107].

#### 3.2.6. Vaccine Dose and Schedule

Across multiple studies, the dose and schedule of HBVacs were reported to be associated with differences in immunogenicity, seroprotection, and long-term antibody persistence in both infants and adults. In children, seroconversion and antibody titers consistently increased with each vaccine dose. Chai et al. reported that a full 3-dose HepB series achieved 100% seropositivity 11–90 days post-vaccination, with antibody levels remaining above protective thresholds for one year [48]. Qiu et al. demonstrated that a 10 µg vaccine dose yielded the highest post-booster seropositivity and GMT, with stronger pre-booster anti-HBs associated with longer-lasting responses [67]. Higher doses (20 µg) were particularly effective in high-risk infants born to HBsAg-positive mothers, improving antibody levels at seven and 12 months and reducing occult HBV infection compared with 10 µg doses [58,68]. Birth-dose administration was crucial for prevention, with Dhouib et al. showing that vaccination at birth achieved 99.4% efficacy in preventing HBV infection compared with 77% for a first dose at 3 months [71]. Preterm infants and infants with low birth weight also achieved protective antibody levels with 3–4 dose schedules [94]. PVST and revaccination further improved protection in infants who initially failed to mount a sufficient response [57,62]. For infants at high risk due to maternal HBsAg positivity or high viral load, combining the hepatitis B vaccine with HBIG prevented MTCT effectively, with 100 IU doses generally sufficient [49,52,84]. In infants, higher vaccine doses (20 µg) increased high antibody responses and reduced low-response rates, although PMTCT rates were similar between 10 µg and 20 µg doses [58,68].

In adults, some studies reported that higher antigen doses and alternative schedules were associated with higher seroconversion rates and antibody titers. Two studies reported 98.7–100% seroconversion with a 3-dose schedule (0, 1, 6 months), and a GMC rise from 413.6 to 6799.9 mIU/mL between months three and seven [63,64]. Shortened two-dose schedules were also reported to be effective, showing higher seroprotection compared to standard three-dose vaccines in adults, including those with comorbidities [89]. High-dose boosters (30–60 µg) in non-responders produced seroconversion rates of 83–92% and robust geometric mean titers [66,107].

### 3.3. Vaccines for Viral Hepatitis C

Five studies conducted in Asia, Europe, Africa and South America were identified [108,109,110,111,112]. All of these studies were RCTs and included 616 adult participants (Appendix A).

In three studies evaluating HBV or HCV vaccines in chronic HCV patients, vaccine responsiveness was generally lower than in healthy controls. Yan et al. reported that only 56.6% of chronic hepatitis C (CHC) patients maintained protective anti-HBs titers five years after the standard three-dose HBV vaccination, compared to 70.8% in healthy controls. Booster doses restored immunity in both groups [108]. Similarly, Medeiros et al. found that non-cirrhotic HCV patients had lower seroprotection rates than controls, regardless of whether a standard (20 µg) or double (40 µg) HBVac dose was used, and a fourth booster improved seroconversion among initial non-responders [109]. Abd El-Wahab et al. observed that hyporesponsiveness persisted even after successful HCV treatment with direct-acting antivirals (DAAs), particularly in patients with isolated anti-HBc positivity, with only 5.1% achieving protective anti-HBs levels [110]. Beyond prophylactic HBV vaccination, therapeutic HCV vaccines were evaluated in combination with antiviral therapy.

Two studies investigated the immunogenicity and clinical effects of HCV vaccines in patients with CHC. Colombatto et al. showed that the E1E2-MF59 vaccine enhanced humoral and cellular responses and improved viral clearance when combined with Peg-IFNα2a and ribavirin [111]. Likewise, Weiland et al. demonstrated that a DNA-based HCV vaccine induced robust NS3-specific T-cell responses and transient HCV RNA reductions, with some patients achieving sustained virologic response after subsequent standard therapy [112]. Therefore, these findings indicate that CHC infection is associated with impaired vaccine responsiveness, which can be partially overcome with booster doses or combined immunotherapeutic strategies. It should be noted that current treatment standards using DAAs differ significantly from the regimens used in these earlier studies.

### 3.4. Vaccines for Viral Hepatitis E

Five studies involving 117,199 participants from Poland and China were identified [113,114,115,116,117], consisting of one cross-sectional study and four RCTs (Appendix A).

The cross-sectional study by Grzegorzewska et al. reported 76 hemodialysis patients and found that prior exposure to HEV did not reduce responsiveness to the HBV vaccine. INF-λ3 levels were positively correlated with anti-HBs titers among vaccine responders [113]. Chen et al. studied 126 healthy adults and showed that an accelerated HBV vaccination schedule (0, 7, and 21 days) was immunologically non-inferior to the standard schedule, providing a safe and rapid option for urgent immunization [114]. Su et al., presented long-term immunological insights from a RCT of 4273 adults, revealing a steady decline in naturally acquired anti-HEV immunoglobulin G (IgG) levels among placebo recipients, from a GMC of 0.55 World Health Organization units (WU)/mL at baseline to 0.27 WU/mL at 67 months, with 17% becoming seronegative by month 67 [115]. Modeling indicated that 50% of naturally infected individuals would lose detectable antibodies within 14.5 years. In contrast, vaccine recipients experienced a 16–46-fold increase in antibody levels by month seven, with over 99% remaining seropositive at month 67. Antibody decline after month 19 was gradual and similar across groups receiving different doses. Power-law modeling suggested that immunity induced by the vaccine would remain detectable for at least 30 years, with 82.1–99.4% of vaccines predicted to remain seropositive at that time, demonstrating significantly more durable protection than natural infection [115]. Similarly, Cao et al., studied 120 adults and found that a new HEV vaccine was safe and well-tolerated across all tested doses, leading to the selection of the 30-µg formulation for further development. Solicited local reactions within seven days were lower in vaccine groups (6.7–30%) than in controls (40%, *p* = 0.027), while systemic reactions and lab abnormalities were mild or moderate, with no vaccine-related serious adverse events reported. [116]. The study by Zhang et al. confirmed 86.8% vaccine efficacy over 4.5 years and showed that the HEV vaccine provided lasting immunity with a safety profile similar to the HBV vaccine control. A total of 60 HEV cases were identified: seven in the vaccine group (0.3 cases per 10,000 person-years) and 53 in the control group (2.1 cases per 10,000 person-years), resulting in a vaccine efficacy of 86.8% (95% CI: 71–94). Among initially seronegative participants, 87% retained anti-HEV antibodies for at least 4.5 years. Adverse event frequencies were comparable between groups, indicating a favorable safety profile. Immunogenicity and efficacy data demonstrated that the HEV vaccine induced long-lasting immunity and consistent protection over 4.5 years [117]. Based on these studies, HEV vaccination has been associated with favorable safety profiles, measurable immunogenic responses, and sustained protection across the populations studied.

### 3.5. Hepatitis Vaccines in Human Non-Viral Liver Diseases

Eighteen studies [118,119,120,121,122,123,124,125,126,127,128,129,130,131,132,133,134,135] conducted across North America, Europe, Asia, and Australia evaluated HAVacs and HBVacs in patients with non-viral hepatology conditions, including chronic liver disease (CLD), cirrhosis, non-alcoholic fatty liver disease (NAFLD), end-stage liver disease (ESLD), and liver transplant (LT) recipients. These studies included RCTs, cohort studies, cross-sectional analyses, and other observational cohorts, with a total of 74,784 participants spanning all age groups, from infants to adults. The evidence shows that vaccine immunogenicity is reduced in advanced liver disease, particularly in cirrhosis and post-transplant settings—yet meaningful protection can still be achieved, especially with optimized dosing strategies or adjuvanted formulations (Appendix A).

#### 3.5.1. Hepatitis B Vaccines in Human Non-Viral Liver Diseases

Across multiple studies, HBVacs in non-viral liver disease patients were associated with variable immunogenicity strongly influenced by disease stage, transplant status, and host factors.

In CLD and cirrhosis patients, Horta et al., evaluated vaccine-induced immunity by measuring anti-HBs titers at two months, six months, and 12 months following completion of the vaccination schedule [118]. At two months, the overall serologic response rate was 76.8%, with higher response rates observed in the FENDRIX^®^ group (83.7%) compared with the HBVAXPRO^®^ 40 group (72.4%). Anti-HBs levels declined progressively over time, with overall response rates decreasing to 72.8% at six months and 59.7% at 12 months, indicating a gradual decline in immunity. In a U.S. cohort, Amjad et al. reported significantly higher seroprotection rates with Heplisav-B (63%) compared with Engerix-B (45%) [119]. Heplisav-B was associated with a 2.7-fold higher rate of achieving protective anti-HBs titers in patients with CLD, indicating superior immunogenicity. The study also reported that the two-dose Heplisav-B regimen achieved higher seroprotection than the standard three-dose Engerix-B schedule [119]. In cirrhotic patients, Wigg et al., found limited benefit from high-dose accelerated HBV schedules, though high-dose boosting modestly improved responses among initial non-responders [120]. In an RCT of 56 cirrhotic patients awaiting LT, Bagheri Lankarani et al. reported on the use of adjunctive G-CSF with a double-dose HBV vaccination regimen (40 µg at weeks 0, 4, and 8) to assess immunogenicity [125]. The study found no significant differences in post-vaccination anti-HBs titers between the G-CSF and placebo groups (*p* > 0.05). Although the G-CSF group demonstrated a more rapid initial increase in antibody levels, overall seroprotection was not improved, indicating that G-CSF did not confer a meaningful immunologic advantage [125].

Among LT recipients, several studies have highlighted significant variability in response. Post-transplant immune recovery was often incomplete, with response rates ranging from 11% to 27% in the study by Arslan et al., and up to 25% in the study by Lu et al. [124,132]. Although response rates were low, some responders were able to safely discontinue HBIG and antiviral therapy without experiencing reinfection [121]. In contrast, observational studies by Takaki et al. and Bienzle et al. demonstrated that selected patients, especially those with stronger donor immunity or those receiving adjuvanted vaccines, achieved robust anti-HBs titers, enabling successful HBIG withdrawal [126,129]. According to Wang et al., maintaining anti-HBs levels > 100 IU/L post-transplant prevented de novo HBV infection (DNHB), and achieving > 1000 IU/L pre-transplant predicted sustained post-transplant protection [122]. Additionally, pediatric studies demonstrated good immunogenicity (85–90%) despite immunosuppression, particularly in recipients of anti-HBc-negative grafts [128,131].

In NAFLD, HBV vaccine responses were generally preserved. Seniuta et al. reported that although NAFLD children had slightly higher rates of subprotective titers after the primary series (32% vs. 19% in controls, *p* = 0.07), all NAFLD patients mounted an effective response to a booster dose [123]. After boosting, NAFLD children achieved a GMC of 239 IU/L (median 313 IU/L; range 19–1000), compared with a GMC of 148 IU/L (median 686 IU/L; range 1–1000) in controls, of whom 10 out of 13 responded. These findings suggest that NAFLD does not impair booster responsiveness.

#### 3.5.2. Hepatitis A Vaccines in Human Non-Viral Liver Diseases

Studies have reported that HAVac is associated with higher immunogenicity compared to HBVac in liver disease populations, although immunogenicity appeared reduced in advanced disease stages. In a study by Wigg et al., involving cirrhotic adults, it was found that high-dose accelerated HAV vaccination resulted in stronger early immune responses than standard dosing [120]. This led to a clinically significant improvement (15%) in immunogenicity, faster protection onset, and minimal additional cost.

In both pediatric and adult CLD populations, HAV vaccination has shown overall effectiveness, although the response varies depending on disease severity. Ferreira et al. observed robust immunogenicity in children with CLD, with 76% seroconversion after the first dose and 97% after the second dose, despite lower GMTs compared to controls, and with a favorable safety profile [127]. Similarly, Smallwood et al., found that many adult ESLD patients developed measurable antibody responses, although at lower rates than the 97% seroconversion observed in healthy individuals, supporting the recommendation to vaccinate those with CLD before transplantation [130]. Arguedas et al., also demonstrated the impact of disease stage, with compensated patients showing significantly higher seroconversion rates (71.4% after the first dose; 98% after the second) than those with decompensated cirrhosis (37.1% and 65.7%, respectively). Child-Pugh class was found to predict immunologic response, emphasizing the importance of vaccinating prior to decompensation [134].

Studies have reported reduced immunogenicity in the LT setting. Arslan et al., reported very low seroconversion rates (8–26%) among HAV-seronegative post-transplant recipients, compared to 97% in healthy controls. Only three out of 37 patients (8%) seroconverted at one month, increasing to five out of 26 (19%) at six months and six out of 23 (26%) at seven months. Responders had higher total white blood cell and lymphocyte counts and were further from transplant compared to non-responders. None of the unvaccinated patients seroconverted during follow-up. Seroconversion rates in orthotopic liver transplant (OLT) recipients were lower than those in healthy individuals (*p* = 0.001) or pre-OLT patients with CLD (*p* = 0.001). The HAVac was reported to be well tolerated by all patients [133].

### 3.6. Hepatitis Vaccines in Humans with HIV

Forty-one studies, which included 117,143 participants from North America, Europe, Asia, and Africa, were identified [136,137,138,139,140,141,142,143,144,145,146,147,148,149,150,151,152,153,154,155,156,157,158,159,160,161,162,163,164,165,166,167,168,169,170,171,172,173,174,175,176]. The most common study design was cohort studies (14 studies; equal 34%), followed by RCTs (14 studies; equal 34%), and cross-sectional studies (6 studies; 14.6%). The populations studied included PLWH, HIV-exposed but uninfected children, and healthy controls. Vaccines evaluated included HAV and HBV formulations, in various forms such as standard, high-dose, intradermal, double-dose, adjuvanted, and combination schedules. Considerable heterogeneity was observed in sample size, immunologic status, use of antiretroviral therapy (ART) or highly active antiretroviral therapy (HAART), and vaccine dosing regimens. These differences were taken into account when interpreting immunogenicity outcomes (Appendix A).

#### 3.6.1. Immunogenicity and Durability of Hepatitis A Vaccination in PLWH

##### Primary Vaccine Response

Multiple cohort studies have reported high seroconversion rates after two- or three-dose HAV schedules among PLWH on stable ART, although some studies have noted slightly lower responses compared to HIV-negative controls [138,139]. Recent data has shown nearly complete seroconversion at eight months in adults living with HIV (97%), similar to healthy controls (100%) [136]. Previous RCTs reported higher GMT with three-dose schedules, but the differences in seroconversion were described as small. Non-smoking status and viral suppression were consistently linked to better responses [168]. According to Loutan et al., seroconversion (≥20 mIU/mL) in HIV-positive individuals reached 63.6% at month 1 and 91.7% at month 13. Booster doses significantly increased the GMC from 25.5 to 659.2 mIU/mL. The vaccine showed a favorable safety and tolerability profiles and elicited strong immunogenicity. Although initial antibody responses in the HIV-positive group were lower than in healthy adults (93.8–100%), booster vaccination effectively enhanced immunity [170].

##### Long-Term Protection and Antibody Persistence

Studies reported that the stability of vaccine-induced immunity generally remains high, although some studies have noted that it could be influenced by immunologic factors. A large Taiwanese cohort maintained long-term serprotection in 90–97% of individuals at 60 months, with higher peak titers and slower decay predicting persistence [139]. In contrast, a Spanish cohort highlighted a clinically relevant decrease: approximately 20% of vaccinated PLWH failed to maintain protective titers, and a small proportion experienced delayed HAV infection, particularly among men who have sex with men (MSM) during community outbreaks [138]. These findings underscore the need for periodic monitoring in individuals with historical nonresponse, higher BMI, low nadir CD4 counts, or detectable HIV RNA at vaccination.

##### Revaccination and Response to Boosting

Revaccination consistently resulted in rapid and high serologic recovery. In a matched case–control study, patients who were revaccinated achieved up to 98% seroresponse by week 48, surpassing non-revaccinated controls. Booster doses also improved responses in earlier trials, confirming that most individuals with declining antibodies retain immunologic memory [137,142,151].

#### 3.6.2. Immunogenicity and Effectiveness of Hepatitis B Vaccination in PLWH

##### Responses to Standard-Dose Vaccination

Across studies, seroprotection after the standard 3-dose HBV vaccination was reported to be lower in PLWH compared to HIV-uninfected individuals. Response rates typically ranged from 30% to 60% [137,142,143,144,145,146,147,148,150,158,159]. Predictors of nonresponse included lower CD4 counts, detectable HIV RNA, male sex, immune activation, and viremia [141,152,154,155,156]. Children and adolescents living with HIV also showed reduced responses, but ART-mediated viral suppression consistently improved seroconversion rates [137,140,141,144,146,154,157,159,176]. This improvement was not observed in participants receiving HAART [163,164,167,171].

##### High-Dose and Intensified Regimens

Studies have reported that high-dose or intensified multi-dose HBV vaccination schedules are associated with higher antibody responses in PLWH. Several trials have shown higher seroconversion rates and a greater proportion of individuals achieving anti-HBs titers greater than 100 mIU/mL with double doses (40 µg) or four-dose regimens [142,143,144,145]. High-dose schedules have been shown to increase seroprotection by 20–30% and produce more durable responses [142,143]. Four-dose or double-dose strategies have been particularly effective in individuals with isolated anti-HBc, low baseline CD4 counts, or prior vaccine nonresponse. However, some studies have shown that double-dose revaccination does not provide a significant advantage over the standard dose and does not significantly improve the response rate [153,159].

##### Novel Adjuvanted and Intradermal Vaccines

Recent multicountry randomized trials demonstrated the superiority of CpG-adjuvanted HBV vaccines [137]. In adults with prior HBV vaccine failure, seroprotection exceeded 93% with two-dose and 99% with three-dose CpG regimens, significantly outperforming alum-based comparators. Antibody titers frequently surpassed 1000 mIU/mL, with rapid seroprotection by week 12. Intradermal vaccination also demonstrated promising efficacy in small cohorts, achieving universal seroconversion among PLWH with prior nonresponse [140,173].

##### Revaccination in Prior Non-Responders

Studies have reported that revaccination success rates were high when intensified schedules were used [160,169]. Across studies, 70–90% of initial non-responders ultimately achieved seroprotection following additional double-dose or multi-dose regimens [137,158,160,169,174]. Predictors of successful revaccination included higher CD4 counts, suppressed HIV RNA, and shorter intervals between the primary series and revaccination.

#### 3.6.3. Determinants of Vaccine Response Across Studies

Across HAVac and HBVac, multiple studies identified predictors of immunogenicity. Higher CD4 counts and viral suppression at the time of vaccination strongly correlate with improved immunogenicity [137,140,141,144,146,154,157,159,163,164,167,171,176]. Younger age, non-smoking status, and the absence of metabolic comorbidities were associated with more robust responses [149,154,168]. Prior vaccine nonresponse does not preclude later success, especially when adjuvanted or high-dose HBV schedules were used [137,142]. HIV-related immune activation, including elevated CD8+/CD38+/HLA-DR+ T cells or impaired memory B-cell function, has been linked to weaker HBV responses [155,156,166]. These findings support an individualized approach to vaccination, particularly in patients with immunologic vulnerability.

#### 3.6.4. Safety and Tolerability

Across all the studies included, vaccines for HAV and HBV (standard, high-dose, intradermal, or adjuvanted) were well tolerated. Adverse events were generally mild and localized. No study identified consistent serious adverse events related to the vaccine, and no clinically meaningful impact on HIV viral load or CD4 count was observed [139,140,141,157,158].

## 4. Discussion

This scoping review offers a comprehensive overview of global evidence on hepatitis A–E vaccines across various populations, clinical settings, and geographic regions. Among 166 studies published between 2000 and 2025, vaccines consistently showed strong safety and immunogenicity. However, vaccine response varied depending on underlying liver disease, immunosuppression, HIV infection, and chronic viral co-infections, highlighting public health benefits and identifying gaps that could influence future vaccine policies (Figure 2).

### 4.1. Global Availability and Study Heterogeneity

The studies included in this review were geographically diverse, with the largest contributions from China, Europe, and the United States (Figure 3). However, data from Africa and other HBV-endemic regions remain underrepresented, despite bearing a substantial global burden of chronic hepatitis B infection. The heterogeneity of study designs, including RCTs, cohort analyses, and large population-based surveillance studies, provides a multidimensional understanding of vaccine performance in real-world and experimental settings. This geographic imbalance is particularly relevant given the marked regional variation in HBV genotype distribution, with genotypes B and C predominating in East Asia, genotype D in Europe and the Mediterranean region, and genotypes A and E being most prevalent in Africa. Importantly, the wide range of populations examined, from neonates to elderly adults, and from healthy individuals to transplant recipients, enables a robust comparison of vaccine responsiveness across various levels of baseline immune competency. Future studies from underrepresented regions, particularly Africa, are needed to assess vaccine effectiveness across diverse HBV genotypes and enhance the global generalizability of current findings.

### 4.2. Hepatitis A Vaccines

Hepatitis A affects an estimated 1.4 million people each year. The prevalence of the disease differs widely across regions and is closely linked to socioeconomic conditions and access to safe water [4]. In low-income countries with inadequate sanitation, nearly all individuals show seropositivity for HAV IgG, as most children are exposed to the virus at a very young age, often experiencing few or no symptoms. However, wealthier countries such as the United States have much lower rates of HAV IgG seropositivity. Travel to endemic regions and close contact with infected individuals remain the primary risk factors for contracting HAV. Since the HAVac was introduced in 1996, the incidence of new infections in the U.S. has dropped by more than 90%, even though overall vaccination coverage remains modest [177].

In our study, involving over 27,000 participants, both inactivated and live-attenuated HAVacs demonstrated excellent safety and consistent long-term immunogenicity. Seroprotection rates typically exceeded 95%. Adverse events were uniformly mild, consistent with the established safety profile of HAVacs. These findings reinforce the suitability of HAV vaccination across age groups and support its use in high-risk populations.

The non-inferiority of locally manufactured vaccines (e.g., Havisure™, HAVpur Junior, HAPIBEV™) relative to well-established global products indicates strong reliability of region-specific HAVac programs and suggests that local production may be a viable strategy to reduce cost barriers in low-resource settings [15,40].

Long-term follow-up studies, some lasting more than two decades, offer compelling evidence of persistent protective immunity from both inactivated and live-attenuated HAVac [34,35]. While antibody titers decrease over time, strong immune responses remain robust, indicating that immune memory persists even in individuals with lower sub-protective antibody levels. Decreases in HAV hospitalizations at population level in countries with universal vaccination programs, such as Turkey and Panama, further illustrate the real-world effectiveness of these vaccines [22,37].

Although some formulations, such as Healive^®^ or Avaxim™, generated higher antibody levels, these variations did not lead to clinically significant differences in protection. This aligns with the World Health Organization (WHO) stance that all licensed HAVacs offer similar benefits [178].

### 4.3. Hepatitis B Vaccines

By 2006, approximately two billion people had been infected with HBV, with around 360 million living with chronic HBV infection worldwide [4]. Most chronic cases are in regions where the virus is endemic. Effective prevention is possible through HBV vaccination, which has been available since the 1980s. Through its nationwide newborn vaccination program, Thailand has markedly reduced HBV incidence and associated liver complications, serving as a prominent example of successful HBV control [179].

A main finding of our study is that included studies reported persistent immunogenicity of hepatitis B vaccines across more than 250,000 participants worldwide. Across age groups, standard 3-dose schedules yield over 90% seroconversion, with CpG-adjuvanted vaccines (Heplisav-B) conferring superior early antibody responses. Immune memory persists for at least 20–35 years, even when circulating antibodies decline, as evidenced by rapid anamnestic responses to boosters.

Despite the availability of effective vaccines, over 1.5 million new HBV infections occur each year, leading to more than 820,000 deaths annually from liver cirrhosis and HCC. Vaccination against hepatitis B remains a cornerstone of public health strategies for HCC prevention and is critical to the global effort to eliminate HBV. The WHO aims for 90% vaccine coverage to achieve elimination by 2030, yet current global birth-dose coverage remains low, at just 42% [180]. The 37-year follow-up study in China provides some of the strongest evidence that HBVac reduces the risk of liver cancer. This confirms the long-hypothesized link between neonatal vaccination, decreased chronic HBV infection, and long-term cancer prevention [54]. These findings have significant implications for global health, especially in regions where HBV is prevalent.

The available evidence suggests that birth-dose vaccination, combined with HBIG when indicated, serves as the cornerstone of PMTCT. Multiple studies have demonstrated near-complete prevention of infant HBV infection, even in areas with high maternal viral load, when timely vaccination and appropriate prophylaxis were administered [50,57,58,77]. Notably, findings from Africa suggest that integrating HBV birth-dose immunization within existing HIV PMTCT infrastructures may be an effective implementation strategy [52]. These findings highlight that maternal viral load, rather than HBIG dose, is the primary predictor of transmission risk, and that optimized vaccine schedules and immunological monitoring may further improve outcomes. Despite the high overall efficacy of PMTCT interventions, residual risk remains for infants of highly viremic, HBeAg-positive mothers, underscoring the importance of maternal antiviral therapy, PVST, and revaccination for infants with suboptimal early responses.

Studies have reported differences in immunogenicity across HBVac formulations. Studies report that 3A-HBV and CpG-adjuvanted vaccines elicit higher antibody titers than single-antigen alum-adjuvanted products [46,181]. Higher antigen doses in neonates improved antibody responses, although PMTCT rates did not differ significantly. Shortened vaccine schedules and high-dose boosters were effective in adults and non-responders, supporting flexible immunization strategies tailored to risk profiles.

Although there is currently no vaccine specifically targeting HDV, HBVac provides indirect immunoprotection against HDV infection, as HDV requires HBV for its replication and assembly. By preventing HBV infection, neonatal and catch-up HBV vaccination effectively reduces the risk of HDV co-infection, especially in high-risk populations [182]. Several studies have shown that populations with high HBV vaccination coverage exhibit substantially lower prevalence of HDV infection, underscoring the public health importance of HBV immunization not only for hepatitis B control but also as a preventive measure against HDV [183,184]. These findings highlight the broader benefits of universal HBV vaccination programs in mitigating both HBV- and HDV-related liver disease.

Our findings directly align with the WHO viral hepatitis elimination goals for 2030, which prioritize a 90% reduction in new infections and a 65% reduction in hepatitis-related mortality [180]. The strong and stable immunogenicity of HBVac, particularly when administered as a timely birth dose, underscores vaccination as the cornerstone of elimination efforts. However, persistently low global birth-dose coverage and reduced vaccine responsiveness in high-risk populations, including PLWH, transplant recipients, and patients with advanced liver disease, represent key barriers to achieving these targets [5]. Addressing these gaps through expanded vaccine access, optimized dosing strategies, and integration of hepatitis vaccination into maternal–child health and HIV programs will be critical to accelerating progress toward elimination.

To maximize impact in low- and middle-income countries, vaccination strategies should prioritize scalable, cost-effective approaches. Key measures include strengthening timely hepatitis B birth-dose delivery through integration with existing maternal–child health and HIV prevention services, expanding the use of affordable locally manufactured vaccines, and adopting simplified or accelerated dosing schedules to improve completion rates. Community-based outreach, task-shifting to trained non-physician health workers, and targeted vaccination of high-risk groups may further enhance coverage in hard-to-reach populations. In parallel, enhancing cold-chain infrastructure, surveillance, and post-vaccination monitoring is essential to ensure durable protection and support global hepatitis elimination.

### 4.4. Hepatitis C and E Vaccines

The HCV affects an estimated 3% of the global population, totaling nearly 200 million cases worldwide, with significant geographic variation in prevalence. Acute HCV infection is typically asymptomatic, and fulminant cases are extremely rare and only sporadically reported. The lack of early symptoms has hindered the identification of patients for large-scale studies, making it challenging to fully understand the early phase of infection. HCV usually progresses silently to chronic infection and often remains asymptomatic for years, with clinical manifestations typically appearing only after advanced fibrosis or cirrhosis develops, which may overlap with other liver diseases [185]. Evidence regarding vaccination strategies for HCV remains scarce. While there is no licensed HCV vaccine available, the reviewed studies offer valuable insights into vaccine responsiveness among HCV-infected individuals. CHC has been associated with impaired HBV vaccine responsiveness, even after viral clearance with direct-acting antivirals, suggesting persistent immune dysregulation in HCV-cured individuals. This supports the need for modified dosing schedules or adjuvanted vaccines in this group. Therapeutic HCV vaccine candidates, when combined with antiviral therapy, have generated virus-specific cellular responses and modest improvements in viral clearance [111,112]. However, efficacy remains limited, as these vaccines are still in early investigational stages.

HEV is estimated to cause over 20 million acute infections annually, resulting in approximately 3.4 million symptomatic cases and 70,000 deaths each year. Although reported cases in developed countries are relatively rare, this is likely due to underdiagnosis and the generally self-limiting nature of the infection [186]. HEV is the leading cause of enterically transmitted hepatitis worldwide, responsible for over half of all acute hepatitis cases in endemic regions such as India and China. Currently, there is no specific antiviral treatment for hepatitis E, making the development of an effective vaccine the most practical strategy for preventing and controlling HEV infections. The HEV 239 vaccine has demonstrated high efficacy, strong immunogenicity, and excellent long-term antibody persistence, surpassing natural infection in terms of immune durability [115]. Modeling studies suggest immunity may last for over 30 years, with over 80–99% of recipients remaining seropositive. These findings underscore the potential benefits of integrating HEV vaccination into public health programs, particularly for high-risk populations in Asia and Africa.

### 4.5. Vaccination in Non-Viral Liver Disease

HBV infection is known to cause liver disease progression through multiple pathways, ultimately increasing the risk of malignant transformation. Chronic HBV infection can lead to significant liver fibrosis and may advance to cirrhosis, a major predisposing factor for HCC. Moreover, HBV has direct oncogenic effects that can promote the development of HCC even in individuals without cirrhosis. For patients with cirrhosis, preventing new HBV infections is essential, as such infections can worsen liver function or trigger serious complications [3,122]. Therefore, the literature suggests that HBV and HAV vaccination is particularly relevant for individuals with chronic liver disease, given reports of more severe clinical outcomes following these viral infections in this population [119].

In LT patients, appropriate vaccination remains essential. Several studies highlight that recipients of living-donor liver transplants, in particular, should receive HBV vaccination. Among OLT recipients, favorable vaccine responses have been observed, involving both humoral and cellular immune activation [126]. Patients undergoing OLT for acute HBV infection are considered strong candidates for immunization. In addition, chronic HBV carriers who receive grafts from unrelated donors with high anti-HBs antibody levels may also respond well to HBV vaccination. Evidence suggests that vaccine-induced immunity in these patients can generate robust HBV-specific antibody production as well as measurable cellular immune responses in vitro [124].

Our results highlight variability in vaccine responsiveness across non-viral liver diseases. Immunogenicity declines progressively with worsening liver function, with cirrhosis and post-transplant states demonstrating the lowest seroconversion rates. Adjuvanted vaccines such as Heplisav-B and FENDRIX^®^ consistently outperformed standard formulations, and high-dose or accelerated schedules improved outcomes modestly [118,119]. These findings underscore a key principle: hepatitis vaccination should be administered early in the course of liver disease, ideally before decompensation or transplantation. Children with NAFLD demonstrated preserved booster responsiveness, indicating that metabolic liver disease alone does not preclude effective immunization [123].

HBV vaccination remains immunogenic in non-viral liver disease, though responses decrease substantially with cirrhosis and following liver transplantation. Higher-dose regimens, adjuvanted vaccines, and pre-transplant immunization improve outcomes.

### 4.6. Vaccination in People with HIV

Viral hepatitis is common among PLWH and contributes to long-term liver morbidity [136]. Therefore, active immunization against HBV and HAV is strongly recommended in this population. However, evidence regarding the optimal vaccination strategy for individuals who fail to respond to the standard initial vaccination remains limited. While HBV vaccination is advised for all HIV-positive individuals, the conventional dosing schedule often results in suboptimal immune responses [137]. Alternative approaches, such as higher-dose formulations or extended dosing schedules, have been explored, but outcomes have varied across studies.

Evidence from our study, which included 41 studies, indicates that HIV infection impairs both HAVac and HBVac responsiveness, especially in individuals with low CD4 counts, detectable viral loads, or coinfections [139,140,141,142]. Higher vaccine doses, adjuvanted formulations, and multi-dose booster strategies improve immunogenicity. Importantly, improved responses were seen alongside virologic control, emphasizing the importance of coordinated vaccination and HIV treatment strategies. Recently Schnyder et al., reported that both 2-dose and 3-dose regimens of the HepBCpG vaccine achieved superior seroprotection compared to three doses of HepB-alum vaccine among PLWH who had not responded to prior hepatitis B vaccination [137].

The evidence indicates the following: 1. HAV appears to be effective and generally durable in PLWH, although immunity may wane in a subset, necessitating monitoring in high-risk groups. 2. Standard HBV vaccination may be insufficient for many PLWH, while high-dose, 4-dose, intradermal, and CpG-adjuvanted regimens significantly improve seroprotection. 3. Revaccination strategies are reported to enhance immune responses, particularly when tailored to immunologic status. 4. Effective immunologic control of HIV is consistently reported as a key factor influencing vaccine responsiveness.

### 4.7. Strengths, Limitations, and Future Directions

A major strength of this review is the comprehensive synthesis of data across populations and contexts, supplemented by long-term follow-up evidence rarely available for other vaccine-preventable diseases. However, heterogeneity in assays, dosing strategies, follow-up durations, and population characteristics limits direct comparability and statistical data analysis. An important limitation is the disproportionate emphasis on HAV and HBV relative to HCV and HEV. This imbalance reflects the current state of vaccine development and implementation, as effective vaccines are widely available for HAV and HBV, whereas no licensed HCV vaccine exists and HEV vaccines are approved only in limited geographic regions, resulting in fewer long-term and population-based studies. Data gaps remain for hepatitis D vaccination, HEV vaccine use in pregnancy, and long-term immune outcomes in immunocompromised hosts such as transplant recipients and PLWH. Future research should prioritize: standardized serologic thresholds and assay harmonization; evaluation of adjuvanted vaccines in high-risk groups; long-term HEV vaccine performance outside East Asia; and integration of hepatitis vaccination into broader maternal–child health and chronic disease programs.

## 5. Conclusions

Across hepatitis A, B, and E, this scoping review suggests that the vaccines currently available are generally safe and immunogenic and are associated with substantial population-level protection. In contrast, while multiple candidate vaccines for HCV have been investigated, no licensed prophylactic HCV vaccine is currently available, and the therapeutic vaccine approaches identified in this review have shown limited and variable efficacy to date.

The strongest and most consistent evidence supports universal neonatal hepatitis B vaccination, particularly for the PMTCT, which remains a cornerstone of global HBV control and HCC prevention. While these findings reinforce well-established principles, this review integrates long-term efficacy data across diverse populations and clinical contexts, highlighting persistent gaps in vaccine responsiveness among individuals with cirrhosis, transplant recipients, chronic HCV infection, and PLWH. These observations underscore the need for early-life immunization, optimized dosing strategies, and tailored vaccination approaches in immunocompromised populations. Therefore, this synthesis emphasizes that maximizing the public health impact of hepatitis vaccination will depend not only on vaccine efficacy but also on equitable implementation, targeted policy interventions, and continued innovation to address underserved populations and unmet clinical needs.

## Figures and Tables

**Figure 1 vaccines-14-00049-f001:**
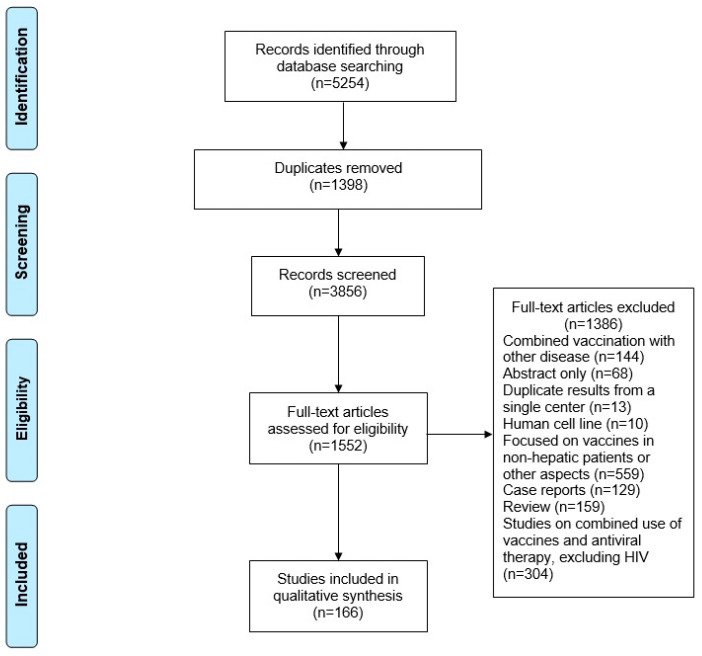
Flow diagram showing the study selection process for the scoping review.

**Figure 2 vaccines-14-00049-f002:**
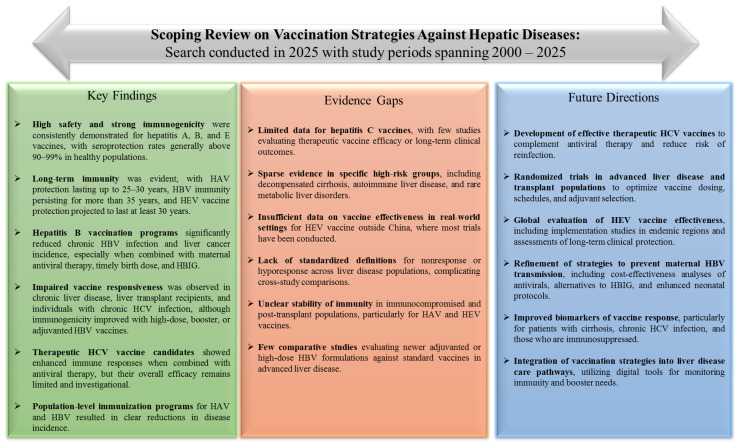
Summary of findings from the scoping review on hepatitis vaccination.

**Figure 3 vaccines-14-00049-f003:**
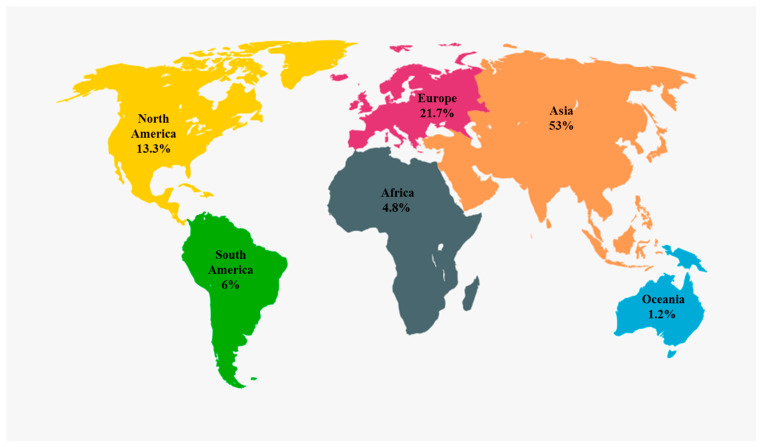
Geographical distribution of the included studies.

## Data Availability

All data will be made available upon request from the corresponding author.

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
