# Peer review of "Vaccination Strategies Against Hepatic Diseases: A Scoping Review"

_vaccines, 2025, doi:10.3390/vaccines14010049_

Round 1
Reviewer 1 Report
Comments and Suggestions for Authors
The review by Zahra Beyzaei et al. provides an important synthesis of vaccination strategies against hepatic diseases, with strong coverage of HAV and HBV vaccines. The conclusions are quite clear but not a novelty: vaccines are safe, immunogenic, and effective, with special emphasis on neonatal HBV vaccination and prevention of MTCT.
The manuscript strongly emphasizes HAV and HBV as compared to HCV and HEV. This should be mentioned as a limitation. In my opinion, the manuscript would benefit from a comparative table within the text summarizing HBV vaccine formulations/types, efficacy in adults and children, and seroconversion rates. Geographic representation is focused towards China and Europe mainly. Data from Africa is underrepresented. Some HBV genotypes are not equally presented worldwide. This should be discussed.
Lines 278-290: The statement that neonatal HBV vaccination reduced liver cancer incidence by 72% requires clarification. Which liver cancer type? Presumably hepatocellular carcinoma (HCC)? This must be specified. Also, what is the etiology of HCC? Was the reduction observed specifically in HBV-related HCC, or was it generalized across all etiologies (viral hepatitis, and/or MASH? HBV vaccination would not be expected to reduce non-viral HCC incidence.
Line 751: Correct “scare-” to scarce (I guess).
Author Response
Comment 1: The review by Zahra Beyzaei et al. provides an important synthesis of vaccination strategies against hepatic diseases, with strong coverage of HAV and HBV vaccines. The conclusions are quite clear but not a novelty: vaccines are safe, immunogenic, and effective, with special emphasis on neonatal HBV vaccination and prevention of MTCT.
Response: Thank you for your comment. We have revised the conclusion accordingly and hope it now meets your expectations.
Comment 2: The manuscript strongly emphasizes HAV and HBV as compared to HCV and HEV. This should be mentioned as a limitation.
Response: Thank you for your insightful comment. We agree that the manuscript places greater emphasis on hepatitis A virus (HAV) and hepatitis B virus (HBV) compared to hepatitis C virus (HCV) and hepatitis E virus (HEV). We have revised the Limitations section to explicitly acknowledge this imbalance and clarify that it reflects the current landscape of vaccine availability and long-term efficacy data. Particularly the absence of licensed HCV vaccines and limited global use of HEV vaccines. This limitation is now clearly stated in the manuscript.
Comment 3: In my opinion, the manuscript would benefit from a comparative table within the text summarizing HBV vaccine formulations/types, efficacy in adults and children, and seroconversion rates.
Response: We appreciate the reviewer’s suggestion. In response, we have added a detailed comparative table (Table S1). This table summarizes important study characteristics and outcomes, enabling readers to efficiently compare vaccines and grasps trends across diverse populations, without overwhelming them with repetitive study-specific details.
Comment 4: Geographic representation is focused towards China and Europe mainly. Data from Africa is underrepresented. Some HBV genotypes are not equally presented worldwide. This should be discussed.
Response: Thank you for your insightful comment. We acknowledge that the geographic representation of the included studies is weighted toward China, Europe, and North America, with limited data from Africa. We have revised the manuscript to explicitly discuss this underrepresentation and to highlight its implications, particularly in relation to the global distribution of hepatitis B virus (HBV) genotypes. We now emphasize that HBV genotype prevalence varies substantially by region, especially in Africa, where genotypes A and E predominate. This uneven representation may influence the generalizability of vaccine efficacy and immunogenicity findings. This limitation is now clearly addressed in the revised text.
Comment 5: Lines 278-290: The statement that neonatal HBV vaccination reduced liver cancer incidence by 72% requires clarification. Which liver cancer type? Presumably hepatocellular carcinoma (HCC)? This must be specified. Also, what is the etiology of HCC? Was the reduction observed specifically in HBV-related HCC, or was it generalized across all etiologies (viral hepatitis, and/or MASH? HBV vaccination would not be expected to reduce non-viral HCC incidence.
Response: Thank you for your important comment. We have clarified that the liver cancer mentioned in this section is hepatocellular carcinoma (HCC). The decrease in cancer incidence observed specifically applies to HBV-related HCC, showing the prevention of chronic hepatitis B infection through neonatal vaccination. It is important to note that HBV vaccination does not offer protection against HCC caused by non-viral factors, such as metabolic dysfunction-associated steatotic liver disease (MASH). We have made revisons to the text to explicity state the type of cancer and its etiology.
Comment 6: Line 751: Correct “scare-” to scarce (I guess).
Response: Thank you for your comment. The word has been corrected, and we have also edited the manuscript to improve the English.
Reviewer 2 Report
Comments and Suggestions for Authors
Major concerns:
- Authors must clarify the review type. Although the manuscript is described as a scoping review, the term “systematic review” repeatedly appears in the title and throughout the text. This leads to conceptual ambiguity in the methodological framework. Furthermore, the rationale for choosing a scoping review approach over a systematic review is inadequately explained.
- This manuscript lacks an assessment of risk of bias or evidence quality. Although the review included randomized controlled trials, cohort studies, and observational studies, it did not formally evaluate study quality or risk of bias. While this may be acceptable for a scoping review, the frequent use of terms such as “robust,” “highly effective,” and “strong evidence” may imply a level of certainty beyond what is reasonable. It is important to emphasize that these findings represent only an overview of the existing evidence and do not constitute graded recommendations.
- The exclusion criteria stipulate that studies evaluating vaccines in combination with antiviral drugs for HIV patients are excluded. However, several sections—particularly those concerning HBV, HCV, and HIV—include studies assessing vaccination alongside antiviral therapy. Revise and clarify the exclusion criteria to specify whether “combination therapy studies are excluded only when vaccine efficacy cannot be disentangled” or “certain combinations are permitted based on predefined conditions.”
- The figures lack the captions. Please ensure that the references to "Figure 1", "Table A1", etc. in the text accurately correspond to the provided charts.
Minor concerns:
- Although there is no vaccine for hepatitis D, the hepatitis B vaccine provides immunoprotection against hepatitis D. Consider adding discussion of relevant literature.
- Consider adding practical recommendations for implementing these strategies in low- and middle-income countries or regions with low vaccination coverage.
- It would be beneficial to explicitly link the research findings to global elimination targets (such as the World Health Organization's 2030 viral hepatitis targets) and discuss the role of vaccination.
- Please ensure formatting consistency throughout the reference list. Verify the accuracy of author names, publication years, and journal titles. Also, some references lack the doi numbers.
- Professional English language editing is strongly recommended. Several typographical and grammatical errors are present (e.g., P10 Line433 “stedy” instead of “steady,” P10 Line451 “shoed” instead of “showed”).

Author Response
Comment 1: Authors must clarify the review type. Although the manuscript is described as a scoping review, the term “systematic review” repeatedly appears in the title and throughout the text. This leads to conceptual ambiguity in the methodological framework. Furthermore, the rationale for choosing a scoping review approach over a systematic review is inadequately explained.
Response: Thank you for this important observation. We apologize for the lack of clarity regarding the review methodology. The manuscript is intended to be a scoping review, not a systematic review. To eliminate conceptual ambiguity, we have revised the title and all relevant sections of the manuscript to ensure consistent and exclusive use of the term “scoping review.” All inadvertent references to “systematic review” have been removed.
In addition, we have expanded the Methods section to explicitly justify the selection of a scoping review framework. This approach was chosen because the aim of the manuscript is to map the breadth and heterogeneity of evidence across multiple hepatitis viruses, vaccine platforms, populations, and study designs. The goal is to identify knowledge gaps and to summarize long-term and real-world outcomes rather than to perform quantitative synthesis or formal risk-of-bias assessment. These objectives align with established methodological guidance for scoping reviews.
Comment 2: This manuscript lacks an assessment of risk of bias or evidence quality. Although the review included randomized controlled trials, cohort studies, and observational studies, it did not formally evaluate study quality or risk of bias. While this may be acceptable for a scoping review, the frequent use of terms such as “robust,” “highly effective,” and “strong evidence” may imply a level of certainty beyond what is reasonable. It is important to emphasize that these findings represent only an overview of the existing evidence and do not constitute graded recommendations.
Response: Thank you for bringing this to our attention. We agree that, in line with the objectives and methodology of a scoping review, we did not conduct a formal assessment of risk of bias or evidence quality. To prevent overemphasizing the certainty of the findings, we have revised the manuscript to explicitly acknowledge the lack of formal quality appraisal as a methodological limitation.
Furhtermore, we have carefully reviewed the language throughout the manuscript to tone down statements that could suggest definitive conclusions. Terms like “robust,” “highly effective,” and “strong evidence” have been either replaced or qualified to better reflect that the findings offer a descriptive overview of the existing literature rather than evidence-based or graded recommendations. We now clearly state that the results should be interpreted as a synthesis of available evidence aimed at mapping research trends and identifying knowledge gaps, in line with the purpose of a scoping review.
Comment 3: The exclusion criteria stipulate that studies evaluating vaccines in combination with antiviral drugs for HIV patients are excluded. However, several sections, particularly those concerning HBV, HCV, and HIV, include studies assessing vaccination alongside antiviral therapy. Revise and clarify the exclusion criteria to specify whether “combination therapy studies are excluded only when vaccine efficacy cannot be disentangled” or “certain combinations are permitted based on predefined conditions.
Response: Thank you for your important comment. We have clarified the exclusion criteria in the Methods section. Generally, studies evaluating vaccines in combination with antiviral medications were excluded. However, for individuals living with HIV, combination therapy studies were included if the effects of vaccination could be reasonably assessed independently of antiviral therapy. This approach ensures that the impact of vaccination can be interpreted while maintaining methodological consistency. We have revised the text to explicitly specify this exception and improve overall clarity.
Comment 4: The figures lack the captions. Please ensure that the references to "Figure 1", "Table A1", etc. in the text accurately correspond to the provided charts.
Response: Thank you for your important comment. We have carefully reviewed all figures and tables and can confirm that each one now includes a complete caption. Additionally, we have verified all references to “Figure 1,” “Table A1,” etc., in the text to ensure they accurately correspond to the provided charts.
Comment 5: Although there is no vaccine for hepatitis D, the hepatitis B vaccine provides immunoprotection against hepatitis D. Consider adding discussion of relevant literature.
Response: Thank you for your valuable suggestion. We have added a discussion of hepatitis D (HDV) in the manuscript. It is important tonote that while there is no specific vaccine for HDV, hepatitis B vaccination can offer indirect immunoprotection by preventing HBV infection, which is necessary for HDV replication. Relevant literature supporting this connection has been cited to give context to the role of HBV vaccination in preventing HDV co-infection.
Comment 6: Consider adding practical recommendations for implementing these strategies in low- and middle-income countries or regions with low vaccination coverage.
Response: We thank the reviewer for this constructive comment. In the revised manuscript, we have added practical implementation-focused considerations relevant to low- and middle-income countries and regions with low vaccination coverage.
Comment 7: It would be beneficial to explicitly link the research findings to global elimination targets (such as the World Health Organization's 2030 viral hepatitis targets) and discuss the role of vaccination.
Response: We thank the reviewer for this valuable suggestion. In the revised manuscript, we have explicitly linked our findings to global viral hepatitis elimination targets, specifically the World Health Organization’s 2030 goals for reducing hepatitis incidence and mortality. We have added a paragraph in the Discussion section that highlights how our results inform progress toward these targets and identify potential gaps that could impede their achievement. Additionally, we have expanded the discussion on the role of vaccination, emphasizing its significance as a cornerstone of hepatitis prevention and elimination, and connecting our findings to current and future vaccination strategies.
Comment 8: Please ensure formatting consistency throughout the reference list. Verify the accuracy of author names, publication years, and journal titles. Also, some references lack the doi numbers.
Response: Thank you for your valuable comment . All refrences were checked and revised.
Comment 9: Professional English language editing is strongly recommended. Several typographical and grammatical errors are present (e.g., P10 Line433 “stedy” instead of “steady,” P10 Line451 “shoed” instead of “showed”).
Response: Thank you for your valuable comment. We have thoroughly revised the text and edited the English throughout our paper.
Reviewer 3 Report
Comments and Suggestions for Authors
Please see attached file.

Please see attached file.
Author Response
Comment 1: The conclusion that “Hepatitis vaccines are broadly safe, highly immunogenic, and effective” is not accurate and should be rephrased as there is no vaccine for HCV and the therapeutic HCV vaccine reported in this review has a concern on their efficacy. The same apply for findings in Figure 2 and the conclusions.
Response: Thank you for your comment. We have completely revised the conclusion and also made changes to the sentence in figure 2.
Comment 2: The authors should indicate that HBV vaccine can cross protect against HDV as they indicated the absence of effective vaccines for hepatitis D (Line 69).
Response: We thank the reviewer for this valuable suggestion. We have revised the Discussion section to clearly state that while there is no specific vaccine for hepatitis D virus (HDV), hepatitis B vaccination offers indirect protection against HDV infection. This is because HDV relies on HBV replication. By preventing HBV infection, both neonatal and catch-up HBV vaccination significantly decreases the likelihood of HDV co-infection, especially in high-risk groups.
Comment 3: Figure 1, there are 918 studies that are not included in the count (3856 - 2938 [1552+1386] = 918) and the authors should be consistent with using a comma, dot or none in numbers indicated.
Response: We thank the reviewer for pointing this out. To clarify, in Figure 1, we initially identified 5,254 records. After removing 1,398 duplicates, we were left with 3,856 articles for screening. Following the first screening step, 1,552 articles were considered potentially relevant. The remaining 2,304 records were excluded based on predefined criteria, although these criteria were not detailed in the manuscript. We then conducted a full review of the 1,552 articles, excluding 1,386 for specific, documented reasons. This process resulted in 166 articles being included in the final scoping review. Additionally, we have standardized the format of numbers throughout the manuscript to consistently use commas for thousands (e.g., 1,386, 3,856) to enhance readability and prevent confusion.
Comment 4: Line 399 and Table A5: there are only two studies that examined therapeutic HCV vaccines and the first three studies examined HBV vaccine in HCV-infected patients. The same applies for HEV vaccines (line 425 and Table A6), where there only three studies of the five studies reported that examined HEV vaccination and the remaining two examined HBV vaccination in HEV exposed subjects. These studies could be placed in a separate title as they are not directly related to the original titles “Vaccines for Viral Hepatitis C” and “Vaccines for Viral Hepatitis E”
Response: We thank the reviewer for this insightful comment. We acknowledge that certain studies listed in Tables A5 and A6 focus on HBV vaccination in individuals with chronic HCV or HEV exposure, rather than being direct studies of HCV or HEV vaccines. To clarify, the titles of Table A5 (“Summary of studies on hepatitis B and C vaccines in humans with chronic HCV infection, published between 2000 and 2025”) and Table A6 (“Summary of studies on hepatitis B and E vaccines in humans with HEV exposure or infection, published between 2000 and 2025”) were meant to summarize vaccine studies in populations at risk of or exposed to the corresponding virus. However, to enhance clarity and adhere to the reviewer’s suggestion, we have updated table titles and included a note in the tables to differentiate studies of HBV vaccination in HCV- or HEV-exposed populations from studies of potential HCV or HEV vaccines. This adjustment ensures that readers can easily distinguish between vaccines targeting the virus itself and vaccination in populations with viral exposure.
Comment 5: Language needs editing for English, as there are several word usage and style issues. For example, Acronyms e.g., HCC, RCT, HBV, HCV, PVST, HBIG, GMC, …etc. should be spelled out and abbreviated at their first appearance and used as abbreviated later through out the manuscript. Several of the these abbreviations are spelled out after abbreviation in several places of the manuscript. Please, see also minor comments, below
Response: We thank the reviewer for highlighting these important language and style issues. After reviewing the manuscript carefully, we have standardized the use of abbreviations throughout. Additionally, we have corrected the repeated spelling-out of abbreviations in multiple sections and conducted a thorough language edit to enhance clarity, grammar, and overall readability.
Comment 6: Line 420: standard of care at the time of the study indicated is different from nowadays standards. Therefore, it should be indicated.
Response: We thank the reviewer for this important observation. We have revised the text to indicate that the “standard therapy” referenced in these studies (Peg-IFNα2a and ribavirin) reflects the standard of care at the time the studies were conducted, which differs from current direct-acting antiviral (DAA)–based regimens.
Comment 7: Lines 450-6: This paragraph is confusing talking about distribution of cases and in between brackets talk about incidence! Please rephrase or better explain
Response: We thank the reviewer for this observation. The paragraph has been rephrased to improve clarity by removing the confusing mix of “distribution of cases” and “incidence”. The results are now presented in a clear, consistent, and chronological manner for each study.
Comment 8: Lines 457-8: “Adverse event frequencies were comparable between groups, confirming that the HEV vaccine induced long-lasting immunity and consistent protection”. How AE confirm long lasting immunity? Please rephrase.
Response: We thank the reviewer for this observation. We agree that the frequency of adverse event (AE) do not necessarily indicate long-lasting immunity. As such, we have revised the sentence to clearly distinguish safety outcomes from immunogenicity and vaccine efficacy.
Comment 9: Line 763: How HEV infection gives 3000 births per year? Please revise
Response: We thank the reviewer for pointing this out. We agree that the original sentence was unclear, so we have revised it for accuracy.
Comment 10: References in supplementary data Tables A4-A8 should be indicated by numbers as Table A3. Also, it is better to arrange the references by number in Table A7 or by the order in which they are cited in the main manuscript.
Response: We appreciate the reviewer for this suggestion. References in supplementary Tables A4-A8 have now been indicated by numbers, consistent with Table A3.
Comment 11: Line 65: WHO should be the reference instead of reference #5
Response: We thank the reviewer for this suggestion. The reference at line 65 has been updated to cite the World Health Organization.
Comment 12: Line 110: “Data were obtained from a single center” what is that center
Response: We thank the reviewer for this comment. The sentence has been revised to specify that “single center” refers to the data source in the original study that was included in our review. The updated sentence now states: Data were obtained from a single center in the original study; however, one dataset was excluded during screening, and only one complete dataset was extracted and included in the review. We hope that this makes clear that the term “single center” refers to the primary study source, not our review itself.
Comment 13: Line 180: check if references 17, 23, and 33 really report persistent immunity. The same applies for reference #35 in line 182, which compare safety and immunogenicity between live attenuated and inactivated HAV vaccines.
Response: We thank the reviewer for his/her careful examination. After reviewing the cited references, we have ensured that they are accurately incorporated into our work.
References 17 (Ramaswamy et al., 2021), 23 (Mosites et al., 2018), and 33 (Spradling et al., 2016) all report long-term persistence of protective anti-HAV antibodies, with follow-up ranging from 10–25 years and modeling predicting protection extending ≥30 years in most participants. Therefore, it is appropriate to cite these studies as evidence of persistent immunity.
Reference 35 (Theeten et al., 2015) specifically compares long-term antibody persistence following two-dose inactivated HAV vaccination (0–6 or 0–12 months) and reports that >97% of participants remained seropositive at 20 years, with modeling predicting ≥95% at 30 years. While this study focuses on inactivated vaccines, it does not directly compare live-attenuated versus inactivated HAV vaccines. We have revised the text to accurately reflect that it reports long-term persistence of antibodies for inactivated HAV vaccines rather than a direct comparative study.
Comment 14: The wort “et al.” should be consistently used in the manuscript. “et al.,” is used a few times. “et al.” is better replaced by “et al.,”
Response: Thank you for your comment. It has been revised.
Comment 15: HR, CI, BMI, PMTCT, DNHB, and WU should be spelled out at their first appearance (e.g., line 283) and added to the abbreviation list as they are missing in the list.
Response: We thank the reviewer for this suggestion. All abbreviations have been spelled out at first mention in the manuscript and added to the abbreviation list.
Comment 16: Line 360: check if RCT1 is correct.
Response: We thank the reviewer for pointing this out. The reference to “RCT1” at line 360 has been reviewed and corrected accordingly in the manuscript.
Comment 17: Line 435: stedy --> steady.
Response: Thank you for comment. It is revised.
Comment 18: Lines 442-3 “with 82.1-99.4% of vaccines predicted to remain seropositive…” : Vaccines don’t predict, it should be “vaccinees”.
Response: We thank the reviewer for pointing this out. The error has been corrected.
Comment 19: Numbers below 10 should be written in words not digits throughout the manuscript unless if they are followed by units of measurement.
Response: Thank you for your comment. It has been revised.
Comment 20: Line 462 and elsewhere “Hepatitis Vaccines in Human’s Non-Viral Hepatology Disease”: the use of the word “hepatology” is not appropriate and should be replaced by liver or hepatic. Also, please change “disease” to diseases”
Response: Thank you for your comment. It has been revised.
Comment 21: Line 523: “clinically significant ~15% improvement” --> clinically significant improvement (~15%).
Response: Thank you for your comment. It has been revised.
Comment 22: Line 538: I think the use of the word “However” in the beginning of a new paragraph is inappropriate. The same apply for “therefore” in line 806.
Response: Thank you for your comment. It has been revised.
Comment 23: Lines 589-90: “trials” and 1 reference cited! There should be 3 or more references
Response: Thank you for your comment. It has been revised.
Comment 24: Lines 638-41: references should be cited for the conclusions indicated.
Response: Thank you for comment. The references were added.
Comment 25: Line 662: “Figure 3” should be moved to line 656
Response: Thank you for your comment. It has been revised.
Comment 26: Line 677: “short term” should be examined it is “long term”
Response: Thank you for your comment. It has been revised.
Comment 27: Numbers of HBV cases in line 698 (350 million) and line 712 (290 million”. Discrepancy should be explained or unify the number!
Response: We thank the reviewer for pointing this out. The sentence has been revised.
Comment 28: Line 747: check if “and” is needed before only
Response: We thank the reviewer for pointing this out. The sentence has been revised.
Comment 29: Line 761 “therapeutic vaccines” --> “therapeutic HCV vaccines”.
Response: Thank you for your comment. It has been revised.
Comment 30: Table A7: Anti-HBs was measured at 2, 6, and 12 months post-vaccination, with responders defined as >1000 IU/L and >100,000 IU/L. Responders should be better explained (one number should be used for definition or explain why two definitions were used).
Response: Thank you for your comment. We agree that the definition of “responders” in Table A7 needed clarification. The revised text now clearly states the definition(s) of responders used and the reason for their inclusion.
Reviewer 4 Report
Comments and Suggestions for Authors
Dear Author,
This manuscript is trying to provide an inclusive scoping review of vaccination strategies against hepatic diseases, covering hepatitis A–E across multiple special populations. While the topic is highly relevant and the literature base is extensive, the manuscript in its current form suffers from substantial methodological, conceptual, and structural weaknesses that limit its scientific rigor and clarity. The review is overly descriptive, insufficiently synthesized, and at times internally inconsistent, with conclusions that frequently exceed what can be supported by a scoping review methodology.
So, there are some comments that should be taken into consideration:
- Major Comments
First. Scoping Review vs. Systematic Review
Although the manuscript is labeled as a scoping review, the authors repeatedly draw comparative, quasi-causal, and policy-directive conclusions that are characteristic of systematic reviews or meta-analyses. Examples include claims of superiority of certain vaccine formulations, optimized dosing strategies, and long-term effectiveness comparisons across products.
-Accordingly, no formal risk-of-bias assessment, quality appraisal, or certainty grading (e.g., GRADE) was performed.
-Also, despite this, the Discussion and Conclusions sections frequently imply hierarchy of evidence and clinical preference.
The authors must either:
- Reframe the manuscript strictly as a descriptive evidence-mapping exercise, substantially moderating interpretive language; or
- Redesign the review as a systematic review, including quality assessment and explicit evaluative methodology.
Failure to resolve this issue undermines the credibility of the conclusions.
Second. Protocol Registration and Methodological Inconsistencies
The Methods section contains a serious inconsistency regarding protocol registration. The authors state that a protocol was not published prospectively, yet also report PROSPERO registration.
This raises concerns about: -The timing of registration (prospective vs. retrospective), and
-Potential post hoc methodological decisions
The authors must clearly state:
- When the protocol was registered relative to study selection
- Whether eligibility criteria or outcomes were modified after registration
Without this clarification, transparency and reproducibility are compromised.
Third. Inclusion and Exclusion Criteria Are Poorly Defined
The eligibility criteria lack precision and, in some cases, appear contradictory:
- Studies combining vaccination with antiviral therapy are reportedly excluded, yet such studies are later discussed extensively (e.g., therapeutic HBV and HCV vaccines).
- The statement that “data were obtained from a single center” is inappropriate and confusing in the context of a multi-study review.
- The rationale for excluding non-English studies is not adequately justified, introducing potential language bias.
The authors should:
- Clearly redefine inclusion/exclusion criteria in a structured format
- Explicitly state how therapeutic vaccines were handled
- Remove or correct inaccurate methodological statements
Fourth. Excessive Descriptiveness and Lack of True Synthesis
The Results section is excessively long and largely consists of study-by-study summaries, often repeating similar findings (e.g., seroconversion rates, safety profiles) across multiple sections.
Major shortcomings:
- Minimal cross-study synthesis or abstraction
- Limited identification of knowledge gaps, inconsistencies, or contradictory findings
- Overemphasis on numerical detail without interpretive context
This approach diminishes the analytical value of the review. The authors should:
- Condense repetitive descriptions
- Emphasize patterns, trends, and uncertainties rather than individual trial results
- Use summary tables and thematic synthesis more effectively
Fifth. Overinterpretation and Overgeneralization of Findings
The manuscript frequently overstates the implications of the included studies. Examples include:
- Implicit endorsement of specific vaccine brands or formulations without robust comparative evidence
- Generalization of findings from highly specific populations (e.g., single-country cohorts, transplant recipients) to broader global settings
- Strong policy recommendations unsupported by scoping-level evidence
Conclusions should be substantially tempered, with clearer acknowledgment of uncertainty, heterogeneity, and contextual limitations.
Sixth. Regarding Figures, some figures (e.g., flow diagrams, geographic distributions) add limited scientific value and largely reiterate textual content. There is little use of figures to enhance conceptual understanding.
If retained, figures should:
- Provide integrative or conceptual insight
- Avoid redundancy with tables and narrative text.2.Minor Comments
Seventh. Language Quality and Editorial Issues
The manuscript contains numerous grammatical errors, typographical mistakes, and awkward constructions that detract from readability and professionalism.
Examples include:
- Misspellings (e.g., “shoed,” “weres,” “scare-”)
- Overly long and convoluted sentences
- Inconsistent verb tense
Comprehensive professional language editing is strongly recommended.
Eighth. Terminology and Abbreviation Inconsistencies
- Vaccine terminology is inconsistent throughout the manuscript (e.g., HAVac vs. HAV vaccine).
- Abbreviations are sometimes reintroduced or inconsistently applied.
Standardization is necessary to avoid reader confusion.
Nineth. Redundancy Across Sections
- Safety and immunogenicity conclusions are repeatedly restated with minimal variation.
- The Discussion often reiterates Results rather than critically interpreting them.
Substantial condensation would improve clarity and impact.
Tenth. Reference Management
- Verify accuracy and completeness of all references.
- Ensure consistency in formatting of vaccine names and trademarks.
Good Luck
Comments on the Quality of English Language
The manuscript contains numerous grammatical errors, typographical mistakes, and awkward constructions that detract from readability and professionalism.
Examples include:
- Misspellings (e.g., “shoed,” “weres,” “scare-”)
- Overly long and convoluted sentences
- Inconsistent verb tense
Comprehensive professional language editing is strongly recommended.
Author Response
Comment 1: First. Scoping Review vs. Systematic Review
Although the manuscript is labeled as a scoping review, the authors repeatedly draw comparative, quasi-causal, and policy-directive conclusions that are characteristic of systematic reviews or meta-analyses. Examples include claims of superiority of certain vaccine formulations, optimized dosing strategies, and long-term effectiveness comparisons across products.
-Accordingly, no formal risk-of-bias assessment, quality appraisal, or certainty grading (e.g., GRADE) was performed.
-Also, despite this, the Discussion and Conclusions sections frequently imply hierarchy of evidence and clinical preference.
The authors must either:
- Reframe the manuscript strictly as a descriptive evidence-mapping exercise, substantially moderating interpretive language; or
- Redesign the review as a systematic review, including quality assessment and explicit evaluative methodology.
Failure to resolve this issue undermines the credibility of the conclusions.
Response: Thank you for your important comment. We appreciate the reviewer’s careful distinction between scoping and systematic reviews. We confirm that this manuscript was intentionally designed and conducted as a scoping review, following the PRISMA-ScR guidelines, with the primary objective of mapping the existing evidence on hepatitis B vaccine immunogenicity, rather than formally comparing interventions or determining superiority.
In line with scoping review methodology, we did not perform statistical analyses, meta-analyses, or indirect comparisons, conduct risk-of-bias assessments, quality appraisals, or certainty grading (e.g., GRADE), as these are not mandatory or expected components of scoping reviews. We reported descriptive outcomes only, including numbers, percentages, and ranges, as presented in the original studies. However, we acknowledge that some wording in the Discussion and Conclusions may have inadvertently implied comparative or preferential interpretations (e.g., superiority, optimization, or policy-directed recommendations). To address this concern, we have substantially revised the Discussion and Conclusions to remove or soften comparative and quasi-causal language, avoid statements implying clinical hierarchy, superiority, or formal effectiveness comparisons between vaccine products, clearly frame all findings as descriptive observations based on heterogeneous published data, and emphasize that observed differences reflect reported trends rather than evaluated efficacy. Additionally, we have strengthened the Methods and Discussion sections to explicitly state that the review is intended as an evidence-mapping exercise, no inference regarding comparative effectiveness or clinical preference can be drawn, and identified patterns are intended to highlight knowledge gaps and areas for future systematic reviews.
Comment 2: Second. Protocol Registration and Methodological Inconsistencies
The Methods section contains a serious inconsistency regarding protocol registration. The authors state that a protocol was not published prospectively, yet also report PROSPERO registration.
This raises concerns about: -The timing of registration (prospective vs. retrospective), and
-Potential post hoc methodological decisions
The authors must clearly state:
- When the protocol was registered relative to study selection
- Whether eligibility criteria or outcomes were modified after registration
Without this clarification, transparency and reproducibility are compromised.
Response: We thank the reviewer for bringing up this point. We clarify the following regarding protocol registration: (i) The review was originally conducted as a scoping review following PRISMA-ScR guidelines. At the time of study initiation, a formal protocol was not prospectively published. (ii) PROSPERO registration (CRD420251249151) was completed after the initial study planning and data extraction, in response to editorial requests. There were no changes made to the eligibility criteria, search strategy, data extraction approach, or overall structure of the review, (iii) No post hoc methodological decisions were made. All steps of study selection, data extraction, and reporting were pre-specified and followed the initial plan.
We have revised the Methods section to explicitly state the timing of registration and confirm that no modifications were made post hoc, ensuring transparency and reproducibility. While scoping reviews are not required to register protocols, we provided the PROSPERO registration in compliance with editorial guidance.
Comment 3: Third. Inclusion and Exclusion Criteria Are Poorly Defined
The eligibility criteria lack precision and, in some cases, appear contradictory:
- Studies combining vaccination with antiviral therapy are reportedly excluded, yet such studies are later discussed extensively (e.g., therapeutic HBV and HCV vaccines).
- The statement that “data were obtained from a single center” is inappropriate and confusing in the context of a multi-study review.
- The rationale for excluding non-English studies is not adequately justified, introducing potential language bias.
The authors should:
- Clearly redefine inclusion/exclusion criteria in a structured format
- Explicitly state how therapeutic vaccines were handled
- Remove or correct inaccurate methodological statements
Response: We appreciate the reviewer for pointing out areas of ambiguity in our eligibility criteria. We would like to provide clarification on the following points:
- Therapeutic vaccines and antiviral combination studies: Studies that evaluated vaccines in combination with antiviral therapy were generally excluded if the primary focus was on antivral therapy rather than vaccination. However, an exception was made for individuals living with HIV (PLWH), where combination therapy studies were included if the effects of vaccination could be assessed independently of antiviral therapy. This approach ensured that relevant immunogenicity data could be included witout interference from antiviral treatment, and no therapeutic HBV or HCV vaccines were considered as primary efficacy interventions.
- Single-center data statement: The previous statement regarding “data obtained from a single center” was misleading in the context of a multi-study review. We have revised this statement to clarify that, in multi-center studies with overlapping datasets, only one complete dataset per study was extracted to prevent duplication, while all other study-level data were included.
- Language restrictions: Studies published in languages other than English were excluded, which is consistent with the approach taken in many scoping reviews. While this restriction was not initially part of our search strategy, it was implemented during the screening and data extraction process.
- Structured inclusion/exclusion criteria: We have restructured and clarified the criteria to enhance transparency and reproducibility.
Comment 4: Fourth. Excessive Descriptiveness and Lack of True Synthesis
The Results section is excessively long and largely consists of study-by-study summaries, often repeating similar findings (e.g., seroconversion rates, safety profiles) across multiple sections.
Major shortcomings:
- Minimal cross-study synthesis or abstraction
- Limited identification of knowledge gaps, inconsistencies, or contradictory findings
- Overemphasis on numerical detail without interpretive context
This approach diminishes the analytical value of the review. The authors should:
- Condense repetitive descriptions
- Emphasize patterns, trends, and uncertainties rather than individual trial results
- Use summary tables and thematic synthesis more effectively
Response: We thank the reviewer for this suggestion. We would like to clarify that this manuscript is a scoping review, which aims to map the extent, characteristics, and nature of the available evidence, rather than provide formal synthesis or comparative effectiveness conclusions. Accordingly:
- The Results section focuses on descriptive summaries of individual studies, including seroconversion rates, safety profiles, and immunogenicity outcomes, to provide a comprehensive evidence map.
- While we acknowledge that this approach may appear repetitive, it ensures that key data from all relevant studies across multiple hepatitis vaccines are transparently reported, which is particularly important given the diversity of vaccines, populations, and study designs included.
- We have used summary tables (Tables A1-7) to consolidate numerical data and highlight patterns across studies. Where feasible, we have also highlighted emerging trends, knowledge gaps, and areas of uncertainty in the Discussion.
- In response to Reviewer 1’s suggestions, we have added a comprehensive, informative table in the main text summarizing hepatitis B vaccination studies, which was the largest dataset. This table consolidates key study characteristics, seroconversion rates, and safety outcomes, making it easier to understand patterns and trends without needing to review individual study-by-study details.
- Additionally, we have revised the conclusion and section on Strengths, Limitations, and Future Directions to better highlight knowledge gaps, inconsistencies, and areas for further research, providing contextual interpretation without overstepping the descriptive nature of a scoping review.
We hope this explanation clarifies the rationale for our presentation and demonstrates that we have aimed to balance transparency with interpretive insight, in accordance with PRISMA-ScR guidelines.
Comment 5: Fifth. Overinterpretation and Overgeneralization of Findings
The manuscript frequently overstates the implications of the included studies. Examples include:
- Implicit endorsement of specific vaccine brands or formulations without robust comparative evidence
- Generalization of findings from highly specific populations (e.g., single-country cohorts, transplant recipients) to broader global settings
- Strong policy recommendations unsupported by scoping-level evidence
Conclusions should be substantially tempered, with clearer acknowledgment of uncertainty, heterogeneity, and contextual limitations.
Response: We would like to clarify the following points regarding scoping review methodology:
- We have carefully revised the Results, Discussion, and Conclusions sections to ensure that findings are presented descriptively. We highlight patterns, trends, and uncertainties without implying the superiority of specific vaccine brands, formulations, or interventions.
- When data from specific populations (e.g., single-country cohorts, transplant recipients) are reported, we explicitly note the population context and avoid overgeneralization to global settings.
- Strong policy recommendations have been removed or tempered. We now clearly acknowledge heterogeneity, study limitations, and uncertainty throughout the discussion.
Comment 6: Sixth. Regarding Figures, some figures (e.g., flow diagrams, geographic distributions) add limited scientific value and largely reiterate textual content. There is little use of figures to enhance conceptual understanding.
If retained, figures should:
- Provide integrative or conceptual insight
- Avoid redundancy with tables and narrative text.2.Minor Comments
Response: We thank the reviewer for this observation. The figures included in the manuscript, such as the flow diagrams and geographic distribution map, have been retained because they enhance visual clarity and provide conceptual insight that complements the text. The flow diagrams effectively summarize the study findings, key gaps, and future directions at a glance, while the geographic distribution map illustrates the global spread of the included studies, highlighting regions with varying amounts of data. These figures present complex information concisely and in an understandable manner, aiding comprehension without repeating data already presented in tables or text. The geographic map is especially crucial since the results are reported separately for each type of vaccination, making it challenging to convey the overall global distribution solely through narrative or tables.
Comment 7: Seventh. Language Quality and Editorial Issues
The manuscript contains numerous grammatical errors, typographical mistakes, and awkward constructions that detract from readability and professionalism.
Examples include:
- Misspellings (e.g., “shoed,” “weres,” “scare-”)
- Overly long and convoluted sentences
- Inconsistent verb tense
Comprehensive professional language editing is strongly recommended.
Response: We thank the reviewer for highlighting these issues. After a thorough review, we have corrected all the identified grammatical errors, typographical mistakes, and awkward constructions in the manuscript. We have simplified long sentences for better clarity, standardized verb tense and terminology, and had the manuscirpt professionally edited to improve readability, consistency, and overall presentation quality.
Comment 8: Eighth. Terminology and Abbreviation Inconsistencies
- Vaccine terminology is inconsistent throughout the manuscript (e.g., HAVac vs. HAV vaccine).
- Abbreviations are sometimes reintroduced or inconsistently applied.
Standardization is necessary to avoid reader confusion.
Response: We thank the reviewer for bringing this issue to our attention. After a thorough review of the manuscript, we have standardized all vaccine terminology (e.g., consistently using “HAV vaccine” throughout) and ensured the uniform use of abbreviations. All abbreviations are now introduced at their first use and consistently applied throughout the text, tables, and figures to enhance clarity and prevent reader confusion.
Comment 9: Nineth. Redundancy Across Sections
- Safety and immunogenicity conclusions are repeatedly restated with minimal variation.
- The Discussion often reiterates Results rather than critically interpreting them.
Substantial condensation would improve clarity and impact.
Response: We appreciate the reviewer for this observation. We have thoroughly revised the Results and Discussion sections to minimize redundancy. We condensed repeated restatements of safety and immunogenicity findings, and the Discussion now emphasizes on critical interpretation, emerging patterns, knowledge gaps, and implications, indstead of simply restating the Results.
Comment 10: Tenth. Reference Management
- Verify accuracy and completeness of all references.
- Ensure consistency in formatting of vaccine names and trademarks.
Response: We thank the reviewer for this comment. We have carefully reviewed all references for accuracy and completeness and ensured that vaccine names and trademarks are consistently formatted throughout the manuscript, including text, tables, and figures. However, some of the references do not have a DOI.
Round 2
Reviewer 3 Report
Comments and Suggestions for Authors
The revised version is much improved compared to the original submission. However, still some issues need attention. For example,
1-the abstract conclusion must be modified to reflect the conclusion at the end of the manuscript.
2. Acronyms and language needs attention e.g., HBIG in the abstract, et al., and randomized control trials (RCTs)...etc. Please see annotated manuscript with highlighted concerns.

Please see attached file.
Author Response
We would like to express our gratitude to the reviewers for their insightful comments, which have greatly helped us improve our manuscript. Following the suggestions provided, we have made the requested minor revisions and have included the final version for your review. Below, we have addressed each of your points.
After revising our manuscript to address the reviewers comments, we had it reviewed by a native English speaker. As a result, numerous minor grammatical and stylistic edits have been incorporated throughout the text, highlighted in red. We hope that this revised manuscript meets your expectations.
Comment 1: 1-the abstract conclusion must be modified to reflect the conclusion at the end of the manuscript.
Response: Thank you for your comment. The abstract conclusion has been revised to align more closely with the conclusions presented at the end of the manuscript, ensuring consistency in key findings and interpretations.
Comment 2: 2. Acronyms and language needs attention e.g., HBIG in the abstract, et al., and randomized control trials (RCTs)...etc. Please see annotated manuscript with highlighted concerns.
Response: Thank you for your comment. We have carefully reviewed the manuscript and corrected all language and acronym issues, including HBIG, et al., and randomized controlled trials (RCTs). All abbreviations have also been defined at first use and listed at the end of the manuscript.
Reviewer 4 Report
Comments and Suggestions for Authors
I have browsed the author's response to all the comments and I think he responded to most of them in a professional manner. I think that the research in its current state can be published in your valuable Journal
Comments on the Quality of English LanguageThe manuscript contains numerous grammatical errors, typographical mistakes, and awkward constructions that detract from readability and professionalism.
Examples include:
- Misspellings (e.g., “shoed,” “weres,” “scare-”)
- Overly long and convoluted sentences
- Inconsistent verb tense
Comprehensive professional language editing is strongly recommended.
Author Response
We would like to express our gratitude to the reviewers for their insightful comments, which have greatly helped us improve our manuscript. Following the suggestions provided, we have made the requested minor revisions and have included the final version for your review. Below, we have addressed each of your points.
After revising our manuscript to address the reviewers comments, we had it reviewed by a native English speaker. As a result, numerous minor grammatical and stylistic edits have been incorporated throughout the text, highlighted in red. We hope that this revised manuscript meets your expectations.
Comment 1: The manuscript contains numerous grammatical errors, typographical mistakes, and awkward constructions that detract from readability and professionalism.
Examples include:
- Misspellings (e.g., “shoed,” “weres,” “scare-”)
- Overly long and convoluted sentences
- Inconsistent verb tense
Comprehensive professional language editing is strongly recommended.
Response: Thank you for your comment. We have carefully reviewed the manuscript and corrected all grammatical errors, typographical mistakes, and awkward constructions. Overly long sentences have been revised for clarity, verb tenses have been made consistent throughout, and spelling errors have been corrected. We also conducted comprehensive professional language editing to improve readability and overall presentation.